# Preserving Gradient Harmony: A Rotation-Based Gradient Balancing for Multi-Task Conflict Remedy

## Abstract

Multi-task learning (MTL) enables knowledge sharing across tasks but often suffers from gradient conflicts, leading to performance imbalances among tasks. Existing weighting-based methods attempt to balance the directional conflicts by striving for the optimal weights computed from gradient or loss information. However, those indirect weighting operations face a limited balancing effect, as the gradient's per-dimensional sensitivities are omitted. Alternatively, gradient manipulation methods such as PCGrad, GradDrop, etc., directly control the task gradients to eliminate opposing gradient directions, but their over-aggressive operations potentially harm the gradient properties, leading to suboptimal updates. They are associated with the issues of over-correction, order dependence, and poor scalability in high-dimensional task settings. To overcome these limitations, we propose the Rotation-Based Gradient Balancing (RGB), a novel algorithm that rotates normalized task gradients toward a consensus direction using independently optimized per-task angle corrections. Unlike projections, rotations provide fine-grained control that preserves beneficial gradient components, reduces global conflicts holistically, and implicitly incorporates loss change information for balanced optimization. Empirical results demonstrate the effectiveness and consistency of RGB, achieving state-of-the-art performance in various datasets, where RGB is the first method on the QM9 dataset with 11 tasks to surpass single-task baselines on average, and its performance is consistent across various benchmarks ranging from 3–40 tasks. Moreover, we propose the concept of multi-task equilibrium relationship that is supported by our empirical experiment and inferring the phenomenon of miss-correction angular error. We also provide the theoretical global convergence of RGB to Pareto stationary under standard smoothness assumptions.

## 1 Introduction

Multi-task learning (MTL) is a learning paradigm that allows a single model to simultaneously optimize multiple tasks given a set of shared parameters. It promotes knowledge transfer between tasks to improve the model's generalization and reduce computational overhead without the need to train separate redundant models for practical applications. However, MTL might suffer from the gradient conflict issue, where the task-specific gradients are in opposing directions and improper aggregation can lead to suboptimal updates, causing performance imbalances in which dominant tasks overshadow others.

Existing MTL approaches can be broadly categorized into weighting-indirect balancing and gradient-direct balancing methods to resolve the corresponding issues. The former relies on the loss or gradient information to compute an optimal weight for gradient aggregation, without directly altering the task gradients. Methods include MGDA Sener & Koltun (2018), CAGrad Liu et al. (2021a), NAsh-MTL Navon et al. (2022), Famo Liu et al. (2023), etc. However, dynamic weighting optimization is oriented by first-order information, thus omitting the gradient's per-dimension sensitivities, resulting in an ineffective Hessian and potentially sub-optimizing the convergence.

Alternatively, gradient-based methods resolve the gradient conflict or performance imbalance issue by directly altering the gradient properties. For instance, PCGrad Yu et al. (2020) relies on or-

thogonal projections to subtract the conflicting component of one task's gradient from another in a pairwise and random order before unifying the update direction. GradDrop Chen et al. (2020) probabilistically drops the gradient signs based on their alignment across tasks to reduce their interference during backpropagation. GradVac Wang et al. (2020) enhances the multi-task performance of multilingual models by iteratively adjusting conflicting gradients through pairwise projections based on cosine similarities relative to exponentially moving averaged historical correlations to mitigate negative transfer. RI-PCGrad Meng et al. (2024) integrates rescaling with PCGrad to ensure the consideration of magnitude information after the projection correction. RotoGrad Javaloy & Valera (2021) reduces the gradient conflict by introducing a learning-based rotation parameter to transform the task gradients for alignment. However, the rotation's parameter optimization does not take the retention of task-specific information into account, potentially deviating from the optimal state where the task-specific information is highly retained while reaching the global task's alignment.

Recently, MTL works have been dominated by weighting-indirect balancing methods such as ConsMTL Qin et al. (2025b), PIVRG Qin et al. (2025a), Go4Align Shen et al. (2024), BiLB4MTL Xiao et al. (2025), etc. Although gradient-direct balancing methods are currently less prominent, they offer several advantages over traditional loss-weighting approaches in multi-task learning. By directly manipulating gradients rather than merely scaling them with dynamic weights, these methods provide greater control over training dynamics, allowing for more precise resolution of inter-task conflicts and reduced destructive interference Zhang et al. (2024). However, existing gradient-direct balancing methods still face some challenges that can restrict their applicability and performance, particularly in terms of projection rigor, pairwise adjustment mechanisms, and performance inconsistencies.

Most gradient-direct balancing methods face performance inconsistency issues due to their rigorous correction. For example, in PCGrad, for gradients $g_i$ and $g_j$ with negative cosine similarity (angles exceeding 90°), $g_i$ is updated as $g_i \leftarrow g_i - \frac{g_i^\top g_j}{\|g_j\|^2} g_j$, fully removing the conflicting component and ensuring non-negative alignment. GradDrop might potentially drop important gradient information, while GradVac alters the gradients only when the current cosine similarity dips below an EMA-tracked historical average, which can cause delays in adapting to sudden shifts or evolving task dynamics. This "hard-adjustment" prioritizes geometric orthogonality but overlooks the nuanced effects on loss reduction. For minor conflicts—where cosine similarity is only marginally negative—the complete elimination of the opposing component may over-penalize gradients, potentially discarding beneficial directions that could support overall descent without severely impacting other tasks.

Furthermore, gradient-direct balancing methods are mainly conducted in a pairwise manner with random order before aggregation. For $T$ tasks, this requires processing up to $\binom{T}{2}$ pairs, imposing quadratic computational scaling (e.g., 3 tasks involve 3 pairs; 4 tasks involve 6 pairs), which becomes burdensome in high-task regimes. This pairwise focus optimizes local conflicts but neglects a holistic, global view of the gradient system, potentially missing alignments that could optimize aggregate descent across all tasks. Projections are executed sequentially in a shuffled order to approximate symmetry, yet this introduces order-dependence: earlier adjustments alter subsequent ones, which may bias the final gradient and introduce variability despite randomization. Such locality can limit the methods' ability to resolve complex inter-task interactions, particularly in high-dimensional task settings and unbalanced datasets where certain tasks dominate.

To overcome these issues, we propose a novel gradient manipulation algorithm, namely RGB (Rotation-Based Gradient Balancing), that aims to rotate each task's normalized gradient towards a consensus direction to minimize global gradient conflicts while taking the loss change information into consideration. Each task's gradient is adjusted with its corresponding optimized rotation angle to achieve the global minimum of conflict between tasks.

Unlike projections or random dropouts, rotations allow fine-grained control via independent per-task angles $\alpha \in [0, \pi/2]^T$, optimized to minimize a global objective comprising a conflict term (average pairwise misalignment) and a proximity term (deviation from original gradients). This formulation considers holistic gradient interactions and incorporates loss change information implicitly through the consensus direction, enabling more balanced and efficient optimization.

Our contributions are threefold:

- We introduce a novel gradient manipulation algorithm, namely RGB (Rotation-Based Gradient Balancing), a scalable rotation-based method for gradient balancing in MTL that addresses global conflicts while preserving gradient integrity.

- We propose the concept of multi-task equilibrium relationship 2.4 that is supported by our empirical experiment A.8 and inferring the phenomenon of miss-correction angular error A.9. We also provide the theoretical global convergence of RGB to Pareto stationary under Lipschitz smoothness and Robbins-Monro stepsize conditions A.6.

- We show the effectiveness and consistency of our method in achieving state-of-the-art performance based on empirical experiments with various datasets. Experiments also include stress-testing under high-dimensional settings, where RGB is able to deliver superior performance on both QM9 and CelebA datasets with 11 and 40 tasks respectively 3.

## 2 PRELIMINARIES AND METHODOLOGY

### 2.1 PARETO OPTIMALITY, COMMON DESCENT DIRECTION, AND PARETO STATIONARITY

Existing direct gradient-balancing method are over-aggressive in making correction for gradient-conflict in a pairwise manner. The lack of global multi-tasks view might potentially over-looking the inter-task relationship and distorting the multi-tasks performance. We first begin with the definition of Pareto optimality, common descent direction and Pareto Stationarity before proposing a novel concept of multi-task equilibrium relationship and miss-correction phenomena.

Pareto optimality refers to a point at which neither task can be improved without worsening another, and common descent direction refers to a direction in the unified parameter space that simultaneously reduces all task losses. Pareto stationarity regards as a point where no such common descent direction exists to improve all objectives at once. Their definition are as follows:

We consider a multi-task learning problem where we aim to optimize a vector of objective functions. Let $x \in \mathcal{X}$ denote the shared model parameters (equivalent to $\theta$ in later sections) and $F(x) = (f_1(x), \ldots, f_Z(x))$ denote the vector of task-specific losses (equivalent to $L_1(\theta), \ldots, L_T(\theta)$).

**Definition 2.1** (Pareto Optimality). *A point $x^* \in \mathcal{X}$ is called* Pareto optimal *if there does not exist another feasible point $x \in \mathcal{X}$ such that $f_z(x) \leq f_z(x^*)$ for all $z \in [Z]$ and $F(x) \neq F(x^*)$. The set of all Pareto optimal solutions is called* Pareto set, *and the corresponding objective vectors $\{F(x^*)\}$ are regarded as* Pareto front.

**Definition 2.2** (Common Descent Direction). *A vector $v \in \mathbb{R}^D$ is called a* common descent direction *at a point $\theta$ if it simultaneously decreases all task losses at $\theta$, i.e.,*

$$\langle \nabla L_i(\theta), v \rangle < 0 \quad \forall i \in \{1, \ldots, T\}.$$

**Definition 2.3** (Pareto Stationarity). *A point $x \in \mathcal{X}$ is called* Pareto stationary *if no common descent direction exists for all objectives at $x$. Formally, this holds when*

$$\text{range}(\nabla F(x)^\top) \cap \left(-\mathbb{R}_{++}^Z\right) = \emptyset,$$

where $\nabla F(x) = (\nabla f_1(x), \nabla f_2(x), \ldots, \nabla f_Z(x)) \in \mathbb{R}^{d \times Z}$ is the Jacobian of $F(x)$, and $\mathbb{R}_{++}^Z$ denotes the positive orthant cone. When all $f_z(x)$ are strongly convex, every Pareto stationary point is also Pareto optimal.

For further details on the smoothness and gradient conditions to ensure the stability and convergence of the rotation-based gradient balancing, see Definition A.1 and Assumption A.1 in Appendix. Afterward, we define the multi-task equilibrium relationship as 2.4 which will serve as the core to develop our RGB equation 1.

**Definition 2.4** (Multi-task Equilibrium Relationship). *Let $\{\bar{g}_i\}_{i=1}^T$ denote the normalized task gradients and let $g_{adj}$ be the rotated direction. The optimal rotation angles $\{\alpha_i^*\}_{i=1}^T$ that minimize the mean of loss $\mathcal{L}(\alpha_1, \ldots, \alpha_T)$ induce an* Multi-task Equilibrium Relationship (MER), *characterized by:*

- *Maximally reducing global gradient conflicts (global alignment term),*

- *Minimally distorting global task-specific descent directions (global proximity term),*

- *Ensuring that the averaged direction $v := \frac{1}{T}\sum_i r_i^*$ forms a common descent direction.*

As formalized in Lemma A.1 in Appendix, this pairwise cosine bound ensures that no pair of gradients is excessively misaligned, which is crucial for ensuring a proper balance between gradient alignment and task-specific descent directions. Definition A.2 and Theorem A.2 in Appendix establishes that when the optimization point is not Pareto stationary, the averaged rotated gradient $v$ serves as a strict common descent direction, ensuring that all task losses are improved simultaneously.

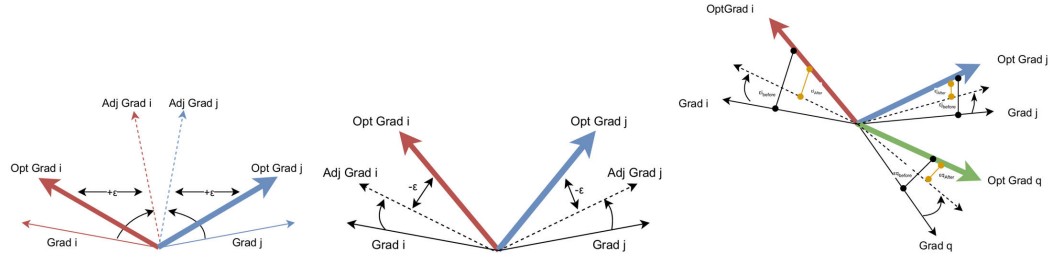

Figure 1: Phenomenon of over-correction (left), under-correction (center) and global miss-correction (right). Denoted: All the task's gradients are normalized in the figure. We also provide an empirical analysis in Appendix A.9.

Besides, the direct gradient balancing methods alter the task gradient based on the subtraction of the task-relative projection Yu et al. (2020); Meng et al. (2024); Wang et al. (2020) or drop of the conflicting gradient Chen et al. (2020) for task-alignment. However, these over-aggressive operating is likely to distort the task-specific descent directions, and the sequential pairwise locally gradient adjustment potentially distort the global conflict level in the gradient system. There is a gap between the previous studies and our definition of the MER 2.4 in terms of global alignment and proximity term. First, we hypothesize the validity of MER 2.4 as a perfect balance between tasks that result in a minimum of the mean loss of tasks. Any gradient adjustment deviates from the MER could result in an imbalance or miss-correction error, sub-optimizing the adjustment and result in unsatisfied performance. As shown in Figure 1, we fine-grain the angular miss-correction scenario into over-correction and under-correction. For simplicity, both scenarios are shown in pairwise manner, where the over-correction happened when the adjusted gradient over-deviated from its theoretical optimal, vise-versa. The global angular miss-correction refers to the sum of the absolute angular over/under-correction error for all the adjusted gradients between optimal gradient as MER 2.4. We define the angular miss-correction, over/under correction error as below.

**Definition 2.5** (Miss-Correction Angular Error (MCAE)). *Let $\alpha_i^*$ denote the optimal rotation angle for task $i$, and $\tilde{\alpha}_i$ be the approximate solution obtained from different gradient-balancing method. The* miss-correction angular error *is defined as*

$$\varepsilon_i := \tilde{\alpha}_i - \alpha_i^*.$$

*We distinguish two cases:*

- *$\varepsilon_i > 0$ (**over-correction**): rotation exceeds the optimal angle.*

- *$\varepsilon_i < 0$ (**under-correction**): rotation falls short of the optimal angle.*

**Definition 2.6** (Over-Correction Angular Error (OCAE)). *Let $\alpha_i^*$ be the theoretically optimal rotation angle that achieves a balance between gradient alignment and task specificity for task $i$. An actual rotation $\tilde{\alpha}_i > \alpha_i^*$ induces an* over-correction, *causing $g_{adj} = r_i(\tilde{\alpha}_i)$ to deviate beyond the optimal point on the alignment–proximity trade-off. This leads to excessive alignment at the cost of task-specific descent fidelity.*

**Definition 2.7** (Under-Correction Angular Error (UCAE)). *Conversely, if the actual rotation angle $\tilde{\alpha}_i < \alpha_i^*$, then $g_{adj} = r_i(\tilde{\alpha}_i)$ remains overly close to its normalized gradient $\bar{g}_i$, insufficiently reducing gradient conflict. This is termed an* under-correction *error. The detailed effect of such over-correction and under-correction can be found in Appendix A.5.*

**Definition 2.8** (Global Miss-Correction Angular Error (GMAE)). *Let $\varepsilon_i$ denote the miss-correction angular error for task $i$. We define the* global miss-correction angular error *(GMAE) and its standard deviation as*

$$\text{GMAE}_t := \frac{1}{T} \sum_{i=1}^{T} |\varepsilon_i^t|, \qquad \text{SD-MAE}_t := \sqrt{\frac{1}{T} \sum_{i=1}^{T} (\varepsilon_i^t - \text{GMAE}_t)^2}.$$

*A high* $\text{SD-MAE}_t$ *implies that some tasks are significantly miscorrected while others are near-optimal, resulting in learning imbalance and unfairness across tasks. For further details, see the analysis in Appendix A.5, where we break down the relationship between miss-correction angular error and the mean task loss.*

## 2.2 ROTATION-BASED GRADIENT BALANCING

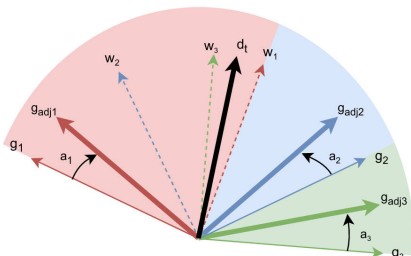

Figure 2: Methodology Illustration of Rotation-based Gradient Balancing

According to the definition of MER 2.4, the ideal task balancing condition is the optimum state between global alignment and global proximity. With this concept, the central of methodology was designed to optimize the ideal rotation angles $\{\alpha_i^*\}_{i=1}^{T}$ that balance the two competing objectives: (i) reducing conflicts among gradients by promoting alignment, and (ii) preserving proximity to the original task gradients so that task-specific information is retained. Afterward, ideal rotation angles were applied to adjust each task gradient toward a shared exponentially moving average (EMA) direction before aggregation. This subsection describes the construction of the rotation operator, the associated optimization objective, and the adaptive procedure that controls the degree of rotation.

Let the shared model parameters be $\theta \in \mathbb{R}^D$ (denoted as $x$ in Section 2.1) and the per-task losses be $L_1(\theta), \ldots, L_T(\theta)$ (denoted as $f_z(x)$ in Section 2.1). For each task, we compute the gradient $g_i = \nabla_\theta L_i(\theta)$ for each task $i$ and normalize it as $\bar{g}_i = \frac{g_i}{\|g_i\|+e}$, where $e$ regards as a small number to avoid the denominator become 0. Afterward, the mean direction of the gradient is computed as $y_t = \frac{1}{T} \sum_{i=1}^{T} \bar{g}_i$, where the cumulative $y_t$ is adopted to compute the exponentially moving average (EMA), which serves as the reference direction for the gradient adjustment. The moving average of the gradient direction is updated as $d_t = \mu d_{t-1} + (1 - \mu) \frac{y}{\|y\|+e}$. Thereafter, it remain essential to define the feasible region for gradient adjustment, in which the ideal rotation angle $\{\tilde{\alpha}_i\}_{i=1}^{T}$ can be optimized within a finite number of optimizations. Intuitively, the adjustment region is bound from angular of $\angle 0°$ to $90°$ from $\bar{g}_i$, where the range initial from $\bar{g}_i$ to the state $w_i$ closest to the reference direction $d_t$. To ensure that the EMA direction converges to a stable mean, we present Proposition A.2, which establishes that the EMA update converges to the smoothed mean direction of the normalized gradients. Then, we compute the normalized vector orthogonal to their normalized gradient $\bar{g}_i$ as $w_i = \frac{d_t - (\bar{g}_i^\top d_t)\bar{g}_i}{\|d_t - (\bar{g}_i^\top d_t)\bar{g}_i\|+e}$.

**Rotation operator** After obtaining $w_i$ and $\bar{g}_i$, the rotation operator is defined to rotate $\bar{g}_i$ toward the EMA direction $d_t$ by optimizing $\tilde{\alpha}_i, \in [0, \pi/2]$ from the objective 1, yielding $g_{\text{adj}} = r_i(\tilde{\alpha}_i) = \cos \tilde{\alpha}_i \bar{g}_i + \sin \tilde{\alpha}_i w_i$, where $w_i$ is orthogonal to $\bar{g}_i$ and aligned with the projection of $d_t$ onto the orthogonal complement of $\bar{g}_i$. This construction ensures that $r_i(\tilde{\alpha}_i)$ interpolates smoothly between the normalized gradient $\bar{g}_i$ (when $\tilde{\alpha}_i = 0$) and the orthogonal component $w_i$ (when $\tilde{\alpha}_i = \frac{\pi}{2}$). Crucially, $w_i$ serves as the reference to define the solution space for the rotation, where selecting the orthogonal vector nearest to $d_t$ ensures the gradient rotates toward the reference direction $d_t$ within

the plane defined by $\bar{g}_i$ and $w_i$. For a formal proof of the interpolation and orthogonal decomposition properties of the rotation operator, refer to Proposition A.1 in the Appendix.

**Objective for rotation angles** Subsequently, our main equation is developed to optimize the optimal rotation angles $\tilde{\alpha}_i, \ldots, \tilde{\alpha}_T$ by minimizing a loss function that combines two terms:

$$\mathcal{L}(\tilde{\alpha}_i) = \underbrace{\frac{1}{T(T-1)} \sum_{i<j} \left[ 1 - r_i(\tilde{\alpha}_i)^\top r_j(\tilde{\alpha}_j) \right]}_{\text{conflict term}} + \underbrace{\lambda \frac{1}{4T} \sum_{i=1}^{T} \left\| r_i(\tilde{\alpha}_i) - \bar{g}_i \right\|^2}_{\text{proximity term}}. \tag{1}$$

The conflict term encourages mutual alignment among rotated gradients by minimizing the average pairwise angular discrepancy, scaled to the $[0, 1]$ interval via the factor $\frac{1}{2}$ (since the original range of $1 - \cos\theta$ is $[0, 2]$). Similarly, the proximity term penalizes deviation from their normalized task gradient $\bar{g}_i$, and is normalized to the $[0, 1]$ range using a factor of $\frac{1}{4}$ (as the squared distance between two opposite unit vectors reaches a maximum of $4$). This normalization ensures that both terms are balanced on a comparable numerical scale, preventing either from dominating the loss due to differences in magnitude and promoting fair contribution to the objective. As guaranteed by Theorem A.1 in Appendix, the rotation objective always has a global minimum, ensuring the existence of an optimal set of rotation angles.

**Complete algorithm** The procedure is summarized in Algorithm 1. Figure 2 illustrates the overview of RGB, where all gradients with the notation $g_i$ in 2 are regarded as normalized gradient $\bar{g}_i$, where $w_i$ is a state that is orthogonal to $\bar{g}_i$ and aligned with the reference direction $d_t$. Afterward, we optimize the objective 1 to approximate the $\tilde{\alpha}_i$ in a global view of the gradient system based on the feasible region formed by $\bar{g}_i$ and $d_t$ before applying them to the rotation operator 2.2 to adjust $\bar{g}_i$ to $g_{\text{adj}}$. Finally, we aggregate all $g_{\text{adj}}$ with equal weighting to form a shared descent direction, before applying it to update the model parameters. The global convergence of Algorithm 1 to a Pareto stationary point is formally established in Appendix A.3.

---

**Algorithm 1** Rotation-Based Gradient Balancing

1: **Inputs:** task losses $L_1, \ldots, L_T$, hyper-parameters ($\mu, \lambda, \epsilon$, update_interval, $\alpha_{\min}, \alpha_{\max}, k_{\text{std}}$)
2: **State:** $\theta \in \mathbb{R}^D$, EMA direction $d_{t-1}$
3: Compute task gradients $g_i = \nabla_\theta L_i(\theta)$ and normalise gradients $\bar{g}_i = g_i/(\|g_i\| + e)$
4: Compute mean direction $y = \frac{1}{T} \sum_{i=1}^{T} \bar{g}_i$
5: Update EMA direction $d_t = \mu d_{t-1} + (1 - \mu)y/(\|y\| + e)$
6: **if** $t \bmod \text{update\_interval} = 0$ **then**
7:  Compute feasible orthogonal corrections state $w_i = \frac{d_t - (\bar{g}_i^\top d_t)\bar{g}_i}{\|d_t - (\bar{g}_i^\top d_t)\bar{g}_i\| + e}$
8:  Initialise $\alpha_i = 0$ and optimize Eq. 1 for $\alpha_{\text{steps}}$ iterations over $\alpha \in [0, \pi/2]^T$,
   where $g_{\text{adj}} = r_i(\alpha_i) = \cos\alpha_i\,\bar{g}_i + \sin\alpha_i\,w_i$
9:  Set final rotated gradients $r_i \leftarrow r_i(\alpha_i^\star)/(\|r_i(\alpha_i^\star)\| + e)$
10: **else**
11:  No rotation $r_i \leftarrow \bar{g}_i$
12: **end if**
13: Compute shared update direction $v = \frac{1}{T} \sum_{i=1}^{T} r_i$,   update $\theta \leftarrow \theta - \eta_t v$

---

**Adaptive control of alpha steps.** The optimization of our equation 1 is based on the stochastic-gradient descent, where its update interval or number of steps namely $\alpha_{\text{steps}}$ is crucial to representing the freedom of search. If higher $\alpha_{\text{steps}}$ is allowed, it could increase the risk of over-correction, leading $g_{\text{adj}}$ to over-deviate from its optimal state hypothesized by 2.4. In contrast, lower $\alpha_{\text{steps}}$ increase the risk of under-correction, leading $g_{\text{adj}}$ to under-deviate from its optimal state. The gap between $g_{\text{adj}}$ with optimal state reflects the miss-correction angular error 2.8, potentially distort the task-specific information or serious trade-off effect, causing the imbalance of multi-task performance. As the veil of $\alpha_{\text{steps}}$ remained as an obstacle, we attempted to adapt a dynamic $\alpha_{\text{steps}}$ based on the variability of task losses. Let the current and previous losses be $L_i^t$ and $L_i^{t-1}$, respectively. Define the relative loss

change and its standard deviation:

$$\delta_i^t = \frac{L_i^t - L_i^{t-1}}{L_i^{t-1} + e}, \qquad s_t = \text{std}(\delta_1^t, \ldots, \delta_T^t). \tag{2}$$

The number of inner iterations is then set adaptively as $\alpha_{\text{steps}}^t = \alpha_{\min} + (\alpha_{\max} - \alpha_{\min})\frac{s_t}{s_t + k_{\text{std}}}$ where $\alpha_{\min}$ and $\alpha_{\max}$ bound the search steps and $k_{\text{std}} > 0$ controls the sensitivity to loss variability.

**Remark 2.1.** *The number of rotation sub-steps $\alpha_{steps}^t$ controls the precision of the approximate solution $\tilde{\alpha}_i^t$. When the per-task loss changes vary significantly (high $\text{SD}(\delta^t)$ in Eq. 2), it often correlates with high $\text{SD-MAE}_t$, indicating task imbalance. In this case, increasing $\alpha_{steps}^t$ improves the approximation of $\alpha_i^*$ for all tasks, thereby reducing both $\text{GMAE}_t$ and $\text{SD-MAE}_t$. This promotes fair learning by ensuring more uniform correction quality across tasks.*

## 3 RESULTS AND DISCUSSION

### 3.1 EVALUATION METRICS

We evaluate our proposed method on several multi-task learning benchmarks, including NYUv2 Silberman et al. (2012), Cityscapes Cordts et al. (2016), QM9 Blum & Reymond (2009), and CelebA Liu et al. (2015). These datasets cover a range of tasks from computer vision to molecular properties prediction, with task numbers ranging from 3 to 40. To assess performance, we follow standard practices in multi-task learning. For each task $i$, we compute the per-task performance drop $\Delta m_i\%$ relative to the STL baseline: $\Delta m_i\% = \frac{1}{S_i}\sum_{j=1}^{S_i}(-1)^{\delta_j}\frac{M_{m,j} - M_{b,j}}{M_{b,j}} \times 100$, where $S_i$ is the number of metrics for task $i$, $M_{b,j}$ is the STL baseline value for metric $j$, $M_{m,j}$ is the value from the MTL method, and $\delta_j = 1$ if higher values are better for metric $j$ (and 0 otherwise). The overall performance drop is then $\Delta m\% = \frac{1}{T}\sum_{i=1}^{T}\Delta m_i$, where $T$ is the number of tasks. Lower (more negative) $\Delta m\%$ indicates better performance relative to STL. We also report the mean rank (MR) across metrics or tasks, where lower MR signifies superior overall ranking. $\text{MR} = \frac{1}{T}\sum_{i=1}^{T}\frac{1}{M_i}\sum_{j=1}^{M_i}R_{i,j}$ where $M_i$ is the number of metrics for task $i$, and $R_{i,j}$ is the rank of the method on metric $j$ of task $i$ (rank 1 being the best among compared methods).

### 3.2 RESULTS ON NYUv2, QM9, CITYSCAPES AND CELEBA

The NYUv2 dataset Silberman et al. (2012) comprises 1,449 densely annotated RGB-D images of indoor scenes, focusing on three pixel-level tasks: 13-class semantic segmentation (measured by mean Intersection over Union (mIoU) and pixel accuracy), monocular depth estimation (absolute and relative error), and surface normal prediction (mean and median angle distances, and percentages of pixels within $11.25°$, $22.5°$, and $30°$ thresholds). For NYUv2, we followed the publicly available ConsMTL implementation to ensure consistency with prior work.

According to Table 1, RGB with the setting of $\lambda = 0$ and $\alpha_{\text{steps}} = 5$ achieved a compatible result with PIVRG and ConsMTL on $\Delta m\%$ (-5.92 vs. -6.50 and -6.72). In particular, our approach excels in segmentation metrics, suggesting better handling of task-specific features. The lower MR for RGB implied the consistent of RGB with a stable ranking across the metrics.

The QM9 dataset Blum & Reymond (2009); Ramakrishnan et al. (2014) is a quantum chemistry benchmark comprising about 134,000 stable organic molecules from the GDB-17 chemical space, represented as graphs and associated with 11 quantum chemical properties (geometric, energetic, electronic, and thermodynamic). Following prior works such as Nash-MTL Navon et al. (2022) and FAMO Liu et al. (2023), we adopt the neural message passing network (MPNN) architecture introduced by Gilmer et al. (2017), with a sequence of message passing layers and a Set2Set pooling operator Vinyals et al. (2015). Using the standard PyTorch Geometric implementation Fey & Lenssen (2019), we split the dataset into 130,831 molecules for training, 10,000 for validation, and 10,000 for testing, and train for 300 epochs. For QM9, we also followed the publicly available ConsMTL implementation to ensure consistency with prior work.

Table 2 demonstrates that our RGB method achieves the lowest $\Delta m\%$ (-3.7) and MR (2.18), outperforming ConsMTL (23.2) by a significant margin. This suggests our approach is particularly

| Method | Segmentation | | Depth | | Surface Normal | | | | | MR ↓ | $\Delta m\%$ ↓ |
| --- | --- | --- | --- | --- | --- | --- | --- | --- | --- | --- | --- |
| | | | | | Angle Dist. ↓ | | Within $t°$ ↑ | | | | |
| | mIoU ↑ | Pix Acc ↑ | Abs Err ↓ | Rel Err ↓ | Mean | Median | 11.25 | 22.5 | 30 | | |
| STL | 38.30 | 63.76 | 0.6754 | 0.2780 | 25.01 | 19.21 | 30.14 | 57.20 | 69.15 | 11.89 | – |
| LS Kendall et al. (2018) | 39.29 | 65.33 | 0.5493 | 0.2263 | 28.15 | 23.96 | 22.09 | 47.50 | 61.08 | 16.89 | 5.59 |
| SI Ruder (2017) | 38.45 | 64.27 | 0.5354 | 0.2201 | 27.60 | 23.37 | 22.53 | 48.57 | 62.32 | 15.11 | 4.39 |
| RLW Lin et al. (2021) | 37.17 | 63.77 | 0.5759 | 0.2410 | 28.27 | 24.18 | 22.26 | 47.05 | 60.62 | 19.89 | 7.78 |
| DWA Liu et al. (2019) | 39.11 | 65.31 | 0.5510 | 0.2285 | 27.61 | 23.18 | 24.17 | 50.18 | 62.39 | 15.89 | 3.57 |
| UW Kendall et al. (2018) | 36.87 | 63.17 | 0.5446 | 0.2260 | 27.04 | 22.61 | 23.54 | 49.05 | 63.65 | 15.78 | 4.05 |
| MGDA Sener & Koltun (2018) | 30.47 | 59.90 | 0.6070 | 0.2555 | 24.88 | 19.45 | 29.18 | 56.88 | 69.36 | 12.89 | 1.38 |
| PCGrad Yu et al. (2020) | 38.06 | 64.64 | 0.5550 | 0.2325 | 27.41 | 22.80 | 23.86 | 49.83 | 63.14 | 16.33 | 3.97 |
| GradDrop Chen et al. (2020) | 39.39 | 65.12 | 0.5455 | 0.2279 | 27.48 | 22.96 | 23.38 | 49.44 | 62.87 | 15.00 | 3.58 |
| CAGrad Liu et al. (2021a) | 39.79 | 65.49 | 0.5486 | 0.2250 | 26.31 | 21.58 | 25.61 | 52.36 | 65.58 | 11.56 | 0.20 |
| IMTL-G Liu et al. (2021b) | 39.35 | 65.60 | 0.5426 | 0.2256 | 26.02 | 21.19 | 26.20 | 53.13 | 66.24 | 10.89 | -0.76 |
| MoCo Fernando et al. (2023) | 40.30 | 66.07 | 0.5575 | **0.2135** | 26.67 | 21.83 | 25.61 | 51.78 | 64.85 | 10.67 | 0.16 |
| Nash-MTL Navon et al. (2022) | 40.13 | 65.93 | 0.5261 | 0.2171 | 25.26 | 20.08 | 28.40 | 55.47 | 68.15 | 7.89 | -4.04 |
| FAMO Liu et al. (2023) | 38.88 | 64.90 | 0.5474 | 0.2194 | 25.06 | 19.57 | 29.21 | 56.61 | 68.98 | 9.44 | -4.10 |
| FairGrad Ban & Ji (2024) | 39.74 | 66.01 | 0.5377 | 0.2236 | 24.84 | 19.60 | 29.26 | 56.58 | 69.16 | 7.11 | -4.66 |
| BiLB4MTL ($\tau$=1) Xiao et al. (2025) | 38.04 | 63.90 | 0.5402 | 0.2278 | 24.70 | 19.19 | 29.97 | 57.44 | 69.69 | 8.56 | -4.40 |
| Aligned-MTL Senushkin et al. (2023) | 40.82 | 66.33 | 0.5300 | 0.2200 | 25.19 | 19.71 | 28.88 | 56.23 | 68.54 | 7.00 | -4.93 |
| GO4Align Shen et al. (2024) | 40.42 | 65.37 | 0.5492 | 0.2167 | 24.76 | 18.94 | 30.54 | 57.87 | 69.84 | 5.11 | -6.08 |
| PIVRG Qin et al. (2025a) | 39.90 | 65.74 | 0.5365 | 0.2243 | **24.30** | 18.80 | 30.95 | 58.26 | **70.38** | **3.89** | -6.50 |
| ConsMTL Qin et al. (2025b) | 40.33 | 65.32 | 0.5491 | 0.2151 | 24.35 | **18.80** | **31.06** | **58.28** | 70.31 | 4.11 | **-6.72** |
| RGB | **41.93** | **67.56** | **0.5294** | 0.2237 | 24.74 | 19.30 | 29.10 | 56.71 | 69.42 | 4.89 | -5.92 |

Table 1: Results on NYUv2. MR values are recomputed using all methods. Lower is better for ↓, higher for ↑. Our RGB results are averaged over three seeds=0,1,2.

| Method | $\mu$ | $\alpha$ | $\epsilon_{\text{HOMO}}$ | $\epsilon_{\text{LUMO}}$ | $\langle R^2 \rangle$ | ZPVE | $U_0$ | $U$ | $H$ | $G$ | $C_v$ | MR ↓ | $\Delta m\%$ ↓ |
| --- | --- | --- | --- | --- | --- | --- | --- | --- | --- | --- | --- | --- | --- |
| | | | | | | MAE ↓ | | | | | | | |
| STL | 0.067 | 0.181 | 60.57 | 53.91 | 0.502 | 4.53 | 58.8 | 64.2 | 63.8 | 66.2 | 0.072 | 3.91 | 0.00 |
| LS | 0.106 | 0.325 | 73.57 | 89.67 | 5.19 | 14.06 | 143.4 | 144.2 | 144.6 | 140.3 | 0.128 | 13.27 | 177.6 |
| SI | 0.309 | 0.345 | 149.8 | 135.7 | **1.00** | 4.50 | 55.3 | 55.75 | 55.82 | 55.27 | 0.112 | 10.00 | 77.8 |
| RLW | 0.113 | 0.340 | 76.95 | 92.76 | 5.86 | 15.46 | 156.3 | 157.1 | 157.6 | 153.0 | 0.137 | 14.82 | 203.8 |
| DWA | 0.107 | 0.325 | **74.06** | 90.61 | 5.09 | 13.99 | 142.3 | 143.0 | 143.4 | 139.3 | 0.125 | 13.00 | 175.3 |
| UW | 0.386 | 0.425 | 166.2 | 155.8 | 1.06 | 4.99 | 66.4 | 66.78 | 66.80 | 66.24 | 0.122 | 12.00 | 108.0 |
| MGDA | 0.217 | 0.368 | 126.8 | 104.6 | 3.22 | 5.69 | 88.37 | 89.4 | 89.32 | 88.01 | 0.120 | 13.64 | 120.5 |
| PCGrad | 0.106 | 0.293 | 75.85 | 88.33 | 3.94 | 9.15 | 116.36 | 116.8 | 117.2 | 114.5 | 0.110 | 11.36 | 125.7 |
| CAGrad | 0.118 | 0.321 | 83.51 | 94.81 | 3.21 | 6.93 | 113.99 | 114.3 | 114.5 | 112.3 | 0.116 | 12.73 | 112.8 |
| IMTL-G | 0.136 | 0.287 | 98.31 | 93.96 | 1.75 | 5.69 | 101.4 | 102.4 | 102.0 | 100.1 | 0.096 | 11.27 | 77.2 |
| Nash-MTL | 0.102 | 0.248 | 82.95 | 81.89 | 2.42 | 5.38 | 74.5 | 75.02 | 75.10 | 74.16 | 0.093 | 8.64 | 62.0 |
| FAMO | 0.150 | 0.300 | 94.00 | 95.20 | 1.63 | 4.95 | 70.82 | 71.2 | 71.2 | 70.3 | 0.100 | 10.09 | 58.5 |
| FairGrad | 0.117 | 0.253 | 87.57 | 84.00 | 2.15 | 5.07 | 70.89 | 71.17 | 71.21 | 70.88 | 0.095 | 9.00 | 57.9 |
| ConsMTL | 0.115 | 0.202 | 82.69 | **67.58** | 1.61 | 3.33 | 48.84 | 49.04 | 49.07 | 49.63 | 0.077 | 4.73 | 23.2 |
| GO4Align | 0.170 | 0.350 | 102.4 | 119.0 | 1.22 | 4.94 | 53.9 | 54.3 | 54.3 | 53.9 | 0.110 | 9.45 | 52.7 |
| BiLB4MTL ($\tau = \sigma$) | 0.230 | 0.290 | 123.89 | 111.95 | 0.97 | 3.99 | 42.73 | 43.1 | 43.2 | 43.1 | 0.097 | 7.27 | 49.5 |
| RGB | **0.064** | **0.190** | 78.07 | 77.55 | 1.09 | **3.06** | **32.9** | **33.4** | **33.1** | **33.2** | **0.065** | **2.18** | **-3.7** |
| Adaptive RGB | 0.089 | 0.203 | 78.23 | 72.99 | 1.21 | 3.42 | 39.62 | 39.51 | 39.54 | 40.29 | 0.071 | 3.27 | 7.1 |

Table 2: Results on QM9. All metrics are MAE (lower is better). MR is the mean rank across the 11 metrics; smaller is better. Our results are the average of three seeds= 0,1,2.

high-task-count scenarios, reducing variance and improving generalization. On this benchmark, our RGB method was evaluated with $\lambda = 0$ and a fixed step size of 30 as its hyperparameters. Notably, RGB outperformed single-task learning (STL) on 8 out of the 11 regression tasks, highlighting its effectiveness in leveraging cross-task information even under the complex multi-task setting. Furthermore, RGB is the first method to achieve a negative $\Delta m\%$ compared to STL which regarding the state-of-the-art result, demonstrating that, in general, it surpasses the average single-task baseline performance. In addition, our adaptive RGB variant was evaluated with $\lambda = 1$ and an adaptive step size schedule. While it achieved a slightly higher MR (3.27), it still delivered strong results, outperforming STL in most of the tasks and reaching a $\Delta m\%$ of 7.1. This shows that incorporating

adaptivity into the step size provides stable performance across the 11 quantum chemical properties, complementing the fixed-step RGB configuration.

The CelebA dataset Liu et al. (2015) and the Cityscapes dataset Cordts et al. (2016) are two widely used multi-task learning benchmarks. CelebA is a large-scale facial attributes dataset containing over 200,000 celebrity images, each annotated with 40 binary attributes, which can be formulated as 40 binary classification tasks. Cityscapes consists of 5,000 high-quality, pixel-level annotated images of urban street scenes collected from 50 cities, and includes three tasks: semantic segmentation (mIoU), disparity estimation (L1 pixel error), and instance segmentation (MSE). Together, these datasets provide diverse and challenging scenarios for evaluating multi-task learning methods, spanning both fine-grained facial attribute recognition and complex scene understanding.

All experiments were conducted based on publicly available implementations. For the CelebA benchmark, we used the FAMO Liu et al. (2023) codebase, while for the Cityscapes benchmark, we built on the publicly released Align-MTL Senushkin et al. (2023) codebase to ensure compatibility with prior work. For CelebA experiments, since existing papers did not release baseline results, we re-ran the STL configuration ourselves based on FAMO repository Liu et al. (2023) and used it as the reference baseline. To ensure fairness, we applied the same training pipeline to all compared methods, and for consistency, all results reported here correspond to the average of three random seeds=0,1,2..

| Method | Cityscapes (3 tasks) | | | | | Method | CelebA | |
| | mIoU [%] ↑ | L1 [px] ↓ | MSE ↓ | MR ↓ | $\Delta m\%$ ↓ | | MR ↓ | $\Delta m\%$ ↓ |
|---|---|---|---|---|---|---|---|---|
| STL | 66.73 | 10.55 | 0.33 | 6.33 | – | STL | 18.84 | – |
| Baseline:Uniform | 52.98 | 10.89 | 0.39 | 14.67 | 14.30 | LS | 10.06 | -27.90 |
| RLW | 51.26 | 10.25 | 0.41 | 15.67 | 15.58 | SI | 11.48 | -26.82 |
| DWA | 53.15 | 10.22 | 0.40 | 13.67 | 12.81 | RLW | 8.98 | -27.72 |
| Uncertainty | 60.12 | **9.87** | 0.33 | 5.33 | 1.15 | DWA | 11.02 | -28.21 |
| MGDA | 66.72 | 17.02 | 0.33 | 11.00 | 20.45 | UW | 10.02 | -27.51 |
| MGDA-UB | 66.37 | 18.63 | 0.32 | 8.67 | 25.05 | MGDA | 16.03 | -25.34 |
| GradNorm | 57.24 | 10.29 | 0.35 | 11.33 | 5.94 | PCGrad | 8.63 | -28.05 |
| GradDrop | 52.98 | 10.09 | 0.40 | 13.00 | 12.49 | GradDrop | 12.64 | -26.05 |
| PCGrad | 54.06 | 9.91 | 0.38 | 8.67 | 9.36 | CAGrad | 9.64 | -26.71 |
| GradVac | 54.07 | 10.39 | 0.40 | 13.00 | 12.89 | IMTL-G | 8.61 | -28.04 |
| CAGrad | 64.33 | 10.15 | 0.34 | 8.33 | 0.95 | Nash-MTL | 8.58 | -28.35 |
| IMTL | 65.13 | 11.58 | 0.32 | 7.33 | 3.04 | FAMO | 9.44 | -27.20 |
| Nash-MTL | 64.84 | 11.90 | 0.37 | 11.67 | 9.25 | GradVac | 7.91 | -28.59 |
| Align-MTL | 67.06 | 10.63 | 0.33 | 7.33 | -0.02 | PIVRG | 9.28 | -27.39 |
| Align-MTL-UB | 66.07 | 10.54 | 0.32 | 6.00 | -0.35 | ConsMTL | **7.37** | -28.65 |
| RGB | 65.05 | 9.90 | 0.32 | 4.67 | **-1.81** | RGB | 9.26 | **-28.71** |
| Adaptive RGB | **65.58** | 10.04 | **0.329** | **3.33** | -1.66 | Adaptive RGB | 9.14 | -27.36 |

Table 3: Results on Cityscapes (3 tasks). MR is recomputed as the mean rank across the three task metrics (mIoU, L1, MSE). Lower is better.

As shown in Table 3, our RGB variant configured with $\lambda = 1$ and a step size of 50 achieves a $\Delta m\%$ of -1.81 on Cityscapes, while the Adaptive RGB variant, using $\lambda = 0$ with an adaptive step size, further improves performance to the best $\Delta m\%$ (-1.66) and the lowest MR (3.33). This demonstrates that our adaptive strategy is particularly effective in balancing the heterogeneous tasks of semantic segmentation, depth estimation, and surface normal prediction in urban scene understanding. Notably, the reduction in MR indicates that our method not only improves absolute task performance but also consistently achieves a higher relative ranking across all tasks when compared to prior methods. On the CelebA benchmark, which involves 40 concurrent facial attribute classification tasks, our RGB variant achieves the best overall $\Delta m\%$ of **-28.71**, surpassing all baselines, while also maintaining a competitive MR of 9.26. The Adaptive RGB variant performs similarly with a $\Delta m\%$ of -27.36 and achieves a lower MR of 9.14, underscoring the robustness of our method under large-scale multi-task settings. These results confirm that our approach not only reduces average performance degradation but also secures a superior task ranking profile across challenging benchmarks.

### 3.3 DISCUSSION

Across all benchmarks, our method consistently achieves state-of-the-art or near-SOTA performance, often surpassing PIVRG and ConsMTL in terms of $\Delta m\%$ and MR. This is attributed to our novel approach in balancing task-specific and shared parameters while reducing performance variance, as inspired by the provided works but extended with adaptive stepping and regularization. The negative $\Delta m\%$ values indicate generalization benefits from multi-tasking. Furthermore, in the Appendix, we provide empirical evidence supporting the existence of Multi-task Equilibrium Relationships (MER) A.8, as well as the phenomena of over-correction and under-correction A.9. The detailed ablation studies and analysis, including the impact of adaptive step sizes A.7 and gradient manipulation methods A.8, validate these concepts across multiple datasets.

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

# A APPENDIX

## A.1 PRELIMINARIES

**Definition A.1** (Lipschitz-Continuous Gradient). *A differentiable function $f$ has a $\beta$-Lipschitz gradient if $\|\nabla f(x) - \nabla f(y)\| \leq \beta \|x - y\|$ for all $x, y \in \mathbb{R}^D$. This condition ensures that the gradient does not change too abruptly.*

**Assumption A.1** (Smoothness and Stepsizes). *The following standard conditions are assumed:*

- **(A1)** *Each task loss $L_i$ is bounded from below: $\inf_\theta L_i(\theta) > -\infty$. This prevents the optimisation process from diverging.*

- **(A2)** *Each loss has a $\beta$-Lipschitz continuous gradient (with the same constant $\beta$ for all tasks, for simplicity).*

- **(A3)** *The stepsizes $(\eta_t)$ satisfy $\sum_{t=0}^{\infty} \eta_t = \infty$ and $\sum_{t=0}^{\infty} \eta_t^2 < \infty$.*

**Remark A.1.** *Assumption (A3) corresponds to the classical Robbins–Monro condition. The first part ensures that sufficient progress can be made asymptotically, while the second limits the cumulative effect of errors (e.g. due to approximate solutions of the rotation sub-problem), ensuring they do not overwhelm the optimization process.*

## A.2 A FUNDAMENTAL GEOMETRIC BOUND

**Lemma A.1** (Feasible Pairwise-Cosine Bound). *Let $\{u_i\}_{i=1}^T$ be unit vectors in $\mathbb{R}^D$ such that $\sum_{i=1}^T u_i = \mathbf{0}$. Then*

$$\max_{i \neq j} \left( -u_i^\top u_j \right) \geq \frac{1}{T-1}.$$

*In words, if a collection of unit vectors balances to zero, then at least one pair must be separated by a cosine similarity of at most $-\frac{1}{T-1}$. Consequently, any family of unit vectors (common descent direction) $\{v\}$ that satisfies $v_i^\top v_j \geq -\frac{1}{T-1}$ for all $i \neq j$ obeys*

$$\langle v_i, \bar{v} \rangle = \frac{1}{T}\left(1 + \sum_{j \neq i} v_i^\top v_j\right) \geq 0, \qquad \bar{v} := \frac{1}{T}\sum_{k=1}^T v_k.$$

*That is, under the cosine bound, each $v_i$ has a nonnegative alignment with the mean vector $\bar{v}$.*

*Proof.* Let $U = [u_1, \ldots, u_T] \in \mathbb{R}^{D \times T}$ and consider its Gram matrix $G = U^\top U$. $G$ is positive semidefinite. Since $\sum_i u_i = \mathbf{0}$, it follows that $G\mathbf{1} = \mathbf{0}$ and because each $\|u_i\| = 1$; hence

$$0 = \mathbf{1}^\top G \mathbf{1} = \sum_{i=1}^T \|u_i\|^2 + \sum_{i \neq j} u_i^\top u_j = T + \sum_{i \neq j} u_i^\top u_j.$$

Dividing both sides by the number of off-diagonal entries, $T(T-1)$, gives

$$\frac{1}{T(T-1)} \sum_{i \neq j} u_i^\top u_j = -\frac{1}{T-1}.$$

Thus, the *average* pairwise cosine is exactly $-1/(T-1)$, which means that at least one pair must realise $u_i^\top u_j \leq -1/(T-1)$, which proves the first statement. The second statement is a direct algebraic rewrite of the projection formula under the assumed bound. $\square$

## A.3 EXISTENCE OF OPTIMAL ROTATION ANGLES

**Goal.** Show that the inner optimisation $\min_{\alpha \in [0, \pi/2]^T} \mathcal{L}(\alpha)$ is well posed so that a rotation angle *optimal $\alpha^\star$* always exists.

**Theorem A.1** (Existence of a Minimiser). *$\mathcal{L}$ attains a global minimum in its feasible set; i.e., there exists $\alpha^\star \in [0, \pi/2]^T$ such that $\mathcal{L}(\alpha^\star) = \min_\alpha \mathcal{L}(\alpha)$.*

*Proof.* The feasible set $[0, \pi/2]^T$ is compact (closed and bounded). Each map $\alpha_i \mapsto r_i(\alpha_i)$ is continuous, and all primitives used in $\mathcal{L}$ (inner products, norms, finite sums) are continuous. By the Weierstrass extreme value theorem, any continuous function on a compact set achieves its minimum. Therefore, $\mathcal{L}$ has at least one global minimiser $\alpha^\star \in [0, \pi/2]^T$. $\qquad\square$

**Remark A.2.** *The key point is that the optimisation is carried out over a bounded and closed domain. This prevents 'escape to infinity' and guarantees that the cost function cannot decrease indefinitely. Hence, the rotation step of the algorithm is always well-defined.*

### A.4    A Strict Common Descent Direction Whenever Non-Stationary

**Goal.** Demonstrate that if $\theta$ is *not* Pareto stationary, the averaged rotated gradient $v := \frac{1}{T}\sum_i r_i(\alpha_i^\star)$ is a descent direction for *all* tasks simultaneously.

**Definition A.2** (Rotation-Adjusted Direction)**.** *For the optimal angles $\alpha^\star$ of Theorem A.1, define*

$$r_i := r_i(\alpha_i^\star), \quad v := \frac{1}{T}\sum_{i=1}^{T} r_i.$$

**Theorem A.2** (Strict Descent When Non-Stationary)**.** *If $\theta$ is not Pareto stationary, then $\langle g_i, v \rangle > 0$ for every $i = 1, \ldots, T$; hence $-v$ is a* strict *common descent direction.*

*Proof.* **Step 1 – Pairwise-cosine bound for $\{r_i\}$.** Suppose, for contradiction, that there exist $p \neq q$ such that $r_p^\top r_q < -1/(T-1)$. Then by slightly rotating $r_p$ and $r_q$ toward the EMA direction $d_t$, their pairwise cosine increases, while the additional proximity penalty in $\mathcal{L}$ is only $O(\Delta\alpha^2)$. For sufficiently small $\Delta\alpha$, this strictly decreases $\mathcal{L}$, contradicting the optimality of $\alpha^\star$. Hence, $r_i^\top r_j \geq -1/(T-1)$ for all pairs.

**Step 2 – Positive projection onto $v$.** By Lemma A.1, the pairwise bound implies $\langle r_i, \bar{v} \rangle \geq 0$, where $\bar{v} := v/\|v\|$ when $v \neq 0$. Since $\theta$ is not Pareto stationary, it leaves margin for further alignment. This strict non-stationarity ensures the inequality is in fact: $\langle r_i, \bar{v} \rangle > 0$.

**Step 3 – Lifting to un-normalised gradients.** Since $\alpha_i^* < \pi/2$ (the proximity term prevents $90°$ rotations), $\cos\alpha_i^\star > 0$. Observing the decomposition $g_i = \|g_i\|\bar{g}_i$ and $\bar{g}_i = \cos\alpha_i^\star r_i - \sin\alpha_i^\star w_i$, one finds

$$\langle g_i, v \rangle = \|g_i\| \left( \cos\alpha_i^\star \langle r_i, v \rangle - \sin\alpha_i^\star \langle w_i, v \rangle \right).$$

The first term is strictly positive; the second term is $O(\sin\alpha_i)$ and cannot cancel the positivity of the first term because $|\langle w_i, v \rangle| \leq \|v\|$. Hence $\langle g_i, v \rangle > 0$.

**Step 4 – Non-degeneracy of $v$.** If $v = 0$, Step 2 would give $\langle r_i, \bar{v} \rangle$ undefined, contradicting the established strict positivity. Therefore $v \neq 0$ and $-v$ is a strict common descent direction. $\qquad\square$

**Remark A.3.** *Intuitively, the optimal rotation ensures that no pair of gradients is "too negatively aligned," which in turn forces every rotated direction to have a positive component along the averaged direction $v$. This alignment then lifts back to the original gradients $g_i$, proving that $-v$ is a strict common descent direction whenever the point is non-stationary.*

### A.5    Theoretical Analysis

This section develops a theoretical analysis of the proposed RGB method. We begin by analyzing the structure of the rotation operator, which interpolates between task-specific gradients and a shared exponential moving average (EMA) direction. This is followed by a first-order analysis showing that the aggregated update direction $v$ constitutes a descent direction for all tasks under suitable alignment. We then introduce the notion of miss-correction angular error, characterizing how deviations from the optimal rotation angle impact both alignment and proximity objectives. Using a second-order Taylor expansion, we derive a decomposition of the mean task loss into two components: the ideal effect of optimally rotated gradients and the deviation caused by miss-correction. This leads to the definition of the global miss-correction angular error (GMAE), which quantifies fairness imbalance across tasks and motivates the use of adaptive step sizes to minimize its impact.

**Proposition A.1** (Rotation Operator Interpolation via Orthogonal Decomposition.). *Let $\bar{g}_i \in \mathbb{R}^D$ be a unit vector (e.g., a normalized gradient), and let $d_t \in \mathbb{R}^D$ be a reference direction. Define the component of $d_t$ orthogonal to $\bar{g}_i$ as $w_i' = d_t - (\bar{g}_i^\top d_t)\, \bar{g}_i$, and let*

$$w_i = \frac{w_i'}{\|w_i'\| + e}$$

*be its normalized form (with $\varepsilon > 0$ for numerical stability). Then, $w_i$ is orthogonal to $\bar{g}_i$, i.e., $\bar{g}_i^\top w_i = 0$ and $\|w_i\| = 1$ (up to $\varepsilon$). For any angle $\alpha_i \in [0, \frac{\pi}{2}]$, the vector $r_i(\alpha_i) = \cos \alpha_i\, \bar{g}_i + \sin \alpha_i\, w_i$ satisfies:*

1. *$\|r_i(\alpha_i)\| = 1$ (up to the $\varepsilon$ regularization),*

2. *$r_i(0) = \bar{g}_i$,*

3. *$r_i\!\left(\frac{\pi}{2}\right) = w_i$ (i.e. aligned purely with the orthogonal component toward $d_t$).*

*Proof.* Compute the inner product:

$$\bar{g}_i^\top w_i' = \bar{g}_i^\top d_t - (\bar{g}_i^\top d_t)\, \bar{g}_i^\top \bar{g}_i = \bar{g}_i^\top d_t - (\bar{g}_i^\top d_t)\,(\|\bar{g}_i\|^2) = \bar{g}_i^\top d_t - (\bar{g}_i^\top d_t) \cdot 1 = 0.$$

Thus, $w_i'$ is orthogonal to $\bar{g}_i$, and so is $w_i$. The normalization ensures $\|w_i\| = 1$ up to the $\varepsilon$ regularization. For the interpolation, since $\bar{g}_i^\top w_i = 0$ and $\|\bar{g}_i\| = \|w_i\| = 1$, we have:

$$\|r_i(\alpha_i)\|^2 = \cos^2 \alpha_i \|\bar{g}_i\|^2 + \sin^2 \alpha_i \|w_i\|^2 + 2 \cos \alpha_i \sin \alpha_i (\bar{g}_i^\top w_i) = \cos^2 \alpha_i + \sin^2 \alpha_i = 1.$$

When $\alpha_i = 0$, we have $r_i(0) = \cos(0)\, \bar{g}_i + \sin(0)\, w_i = \bar{g}_i$.

When $\alpha_i = \frac{\pi}{2}$, we get $r_i(\frac{\pi}{2}) = \cos(\frac{\pi}{2})\, \bar{g}_i + \sin(\frac{\pi}{2})\, w_i = w_i$. $\qquad\square$

**Proposition A.2** (EMA Update Converges to Smoothed Mean Direction). *Let $y_t = \frac{1}{T} \sum_{i=1}^{T} \bar{g}_i$ be the instantaneous mean of normalized gradients, and define the EMA (exponentially moving average) update*

$$d_t = \mu\, d_{t-1} + (1 - \mu)\, \frac{y_t}{\|y_t\| + e}.$$

*Then under standard assumptions (bounded gradients, $0 \le \mu < 1$), $d_t$ converges in direction to the smoothed (infinite-horizon) average of the $\bar{g}_i$'s. In particular, for large $t$,*

$$d_t \approx (1 - \mu) \sum_{s=0}^{t} \mu^{t-s} \frac{y_s}{\|y_s\| + e},$$

*so $d_t$ reflects the past mean directions with exponentially decaying weights.*

*Proof.* By unrolling the recurrence:

$$d_t = \mu d_{t-1} + (1 - \mu) q_t$$

where $q_t := m_t / (\|m_t\| + \varepsilon)$. Then

$$d_t = \mu^t d_0 + (1 - \mu) \sum_{s=1}^{t} \mu^{t-s} q_s.$$

Since $d_0$ is fixed and $\mu^t \to 0$ as $t \to \infty$ (if $0 \le \mu < 1$), the influence of $d_0$ vanishes. Hence for large $t$,

$$d_t \approx (1 - \mu) \sum_{s=1}^{t} \mu^{t-s} q_s$$

which is a weighted average of past $q_s$ with exponentially decaying weights. That proves the claimed smoothing property. $\qquad\square$

**First-Order Analysis.** For a small step size $\eta > 0$, the change in the loss of task $i$' under the shared update direction $v$ is given by the first-order Taylor expansion:

$$L_i(\theta - \eta v) \approx L_i(\theta) - \eta \langle \nabla_\theta L_i, v \rangle + \mathcal{O}(\eta^2).$$

*Proof.* We aim to analyze how task $i$'s loss $L_i(\theta)$ changes when the parameters are updated in the direction $v$ with step size $\eta > 0$. Define the scalar function:

$$f(\eta) := L_i(\theta - \eta v),$$

which expresses the task loss as a function of the scalar step size $\eta$ in direction $v$.

**Step 1: Apply the first-order Taylor expansion.** Let $f : \mathbb{R} \to \mathbb{R}$ be differentiable at $\eta = 0$. Then, the first-order Taylor expansion of $f$ around $\eta = 0$ is given by:

$$f(\eta) \approx f(0) + f'(0)\,\eta + \mathcal{O}(\eta^2),$$

where $\mathcal{O}(\eta^2)$ denotes second- and higher-order terms that vanish faster than $\eta$ as $\eta \to 0$.

**Step 2: Compute $f(0)$ and $f'(0)$.** We compute:

$$f(0) = L_i(\theta - 0 \cdot v) = L_i(\theta).$$

To compute the derivative $f'(\eta)$, we apply the chain rule:

$$f'(\eta) = \frac{d}{d\eta} L_i(\theta - \eta v) = \nabla_\theta L_i(\theta - \eta v)^\top \cdot \frac{d}{d\eta}(\theta - \eta v).$$

Since $\frac{d}{d\eta}(\theta - \eta v) = -v$, this simplifies to:

$$f'(\eta) = -\nabla_\theta L_i(\theta - \eta v)^\top v.$$

Evaluating at $\eta = 0$ gives:

$$f'(0) = -\nabla_\theta L_i(\theta)^\top v = -\langle \nabla_\theta L_i(\theta), v \rangle.$$

**Step 3: Substitute into the Taylor expansion.** Now substituting back into the Taylor expansion:

$$f(\eta) = L_i(\theta - \eta v) = f(0) + f'(0)\,\eta + \mathcal{O}(\eta^2) = L_i(\theta) - \eta \langle \nabla_\theta L_i(\theta), v \rangle + \mathcal{O}(\eta^2).$$

Thus, the first-order approximation of the loss after an update in direction $v$ with small step size $\eta$ is:

$$L_i(\theta - \eta v) \approx L_i(\theta) - \eta \langle \nabla_\theta L_i(\theta), v \rangle + \mathcal{O}(\eta^2),$$

which completes the proof. $\square$

The directional derivative term $-\langle \nabla_\theta L_i, v \rangle$ quantifies how effective the update is in reducing task $i$'s loss. Because each rotated gradient $r_i^*$ lies between the original $\bar{g}_i$ and the shared EMA direction $d_t$, the resulting $v$ incorporates both task-specific and global descent tendencies. This ensures that $v$ acts as a compromise direction that enables consistent progress across all tasks in non-stationary regimes.

**Effect of Over-Correction and Under-Correction.** Let $\mathcal{L}(\alpha)$ be the rotation loss defined in Eq. 1, and suppose that for each task $i$, the optimal rotation angle $\alpha_i^*$ is a local minimizer of $\mathcal{L}$ in its coordinate direction. Then,

$$\frac{\partial \mathcal{L}}{\partial \alpha_i}\bigg|_{\alpha_i = \alpha_i^*} = 0, \qquad \frac{\partial^2 \mathcal{L}}{\partial \alpha_i^2}\bigg|_{\alpha_i = \alpha_i^*} > 0.$$

Now consider the effect of miss-correction $\varepsilon_i = \tilde{\alpha}_i - \alpha_i^* \neq 0$ on the loss. A Taylor expansion around $\alpha_i^*$ gives

$$\mathcal{L}(\tilde{\alpha}_i) = \mathcal{L}(\alpha_i^*) + \frac{1}{2}\varepsilon_i^2 \cdot \frac{\partial^2 \mathcal{L}}{\partial \alpha_i^2}\bigg|_{\alpha_i = \alpha_i^*} + \mathcal{O}(\varepsilon_i^3),$$

which shows that *any deviation* from $\alpha_i^*$ strictly increases the loss up to second order.

*Proof.* We wish to analyze how the rotation loss $\mathcal{L}(\alpha)$ changes due to a miss-correction angular error $\varepsilon_i := \tilde{\alpha}_i - \alpha_i^* \neq 0$, where $\alpha_i^*$ is a local minimizer of $\mathcal{L}$ with respect to coordinate $\alpha_i$.

**Step 1: Define the scalar slice.** Since we are focusing on a single coordinate $\alpha_i$, define the scalar function

$$f(\alpha_i) := \mathcal{L}(\alpha_1^*, \ldots, \alpha_{i-1}^*, \alpha_i, \alpha_{i+1}^*, \ldots, \alpha_T),$$

which evaluates the global loss by varying only $\alpha_i$, holding the other rotation angles fixed at their optimal values. Thus, $f(\alpha_i)$ is a univariate scalar function that captures how $\mathcal{L}$ behaves along the $\alpha_i$ direction.

**Step 2: Use Taylor expansion around $\alpha_i^*$.** Because $f$ is assumed twice differentiable near $\alpha_i^*$, the second-order Taylor expansion of $f$ at point $\alpha_i^* + \varepsilon_i$ gives:

$$f(\alpha_i^* + \varepsilon_i) = f(\alpha_i^*) + f'(\alpha_i^*) \cdot \varepsilon_i + \frac{1}{2} f''(\alpha_i^*) \cdot \varepsilon_i^2 + \mathcal{O}(\varepsilon_i^3),$$

where $f'(\alpha_i^*)$ and $f''(\alpha_i^*)$ denote the first and second derivatives of $f$ evaluated at $\alpha_i^*$.

**Step 3: Use optimality conditions at $\alpha_i^*$.** Since $\alpha_i^*$ is a local minimizer of $f$ (and hence of $\mathcal{L}$ in coordinate $i$), we have the standard first- and second-order conditions:

$$f'(\alpha_i^*) = 0, \quad f''(\alpha_i^*) > 0.$$

These ensure that $\alpha_i^*$ is a strict local minimum of the function $f$.

**Step 4: Simplify the expansion.** Substituting these values into the Taylor expansion, we obtain:

$$f(\alpha_i^* + \varepsilon_i) = f(\alpha_i^*) + \frac{1}{2} f''(\alpha_i^*) \cdot \varepsilon_i^2 + \mathcal{O}(\varepsilon_i^3).$$

**Step 5: Interpret the result.** This implies:

$$\mathcal{L}(\tilde{\alpha}_i) = \mathcal{L}(\alpha_i^*) + \frac{1}{2} \varepsilon_i^2 \cdot \left. \frac{\partial^2 \mathcal{L}}{\partial \alpha_i^2} \right|_{\alpha_i = \alpha_i^*} + \mathcal{O}(\varepsilon_i^3),$$

where we identify $f(\alpha_i^* + \varepsilon_i) = \mathcal{L}(\tilde{\alpha}_i)$.

Therefore, any miss-correction angular error leads to a second-order increase in the rotation loss $\mathcal{L}$, showing that both over-correction and under-correction degrade optimization quality. $\qquad\square$

More specifically, decompose $\mathcal{L}$ into its alignment and proximity terms in Eq. 1.

- If $\varepsilon_i > 0$ (over-correction), then locally:

$$\frac{\partial}{\partial \alpha_i} \left[ 1 - r_i^\top r_j \right] < 0, \qquad \frac{\partial}{\partial \alpha_i} \| r_i - \bar{g}_i \|^2 > 0.$$

  That is, the alignment improves (conflict decreases), but task specificity degrades.

- If $\varepsilon_i < 0$ (under-correction), then locally:

$$\frac{\partial}{\partial \alpha_i} \left[ 1 - r_i^\top r_j \right] > 0, \qquad \frac{\partial}{\partial \alpha_i} \| r_i - \bar{g}_i \|^2 < 0.$$

  That is, the rotated direction stays closer to the original gradient, but fails to adequately reduce gradient interference.

In both cases, the deviation from $\alpha_i^*$ yields a suboptimal compromise between gradient alignment and proximity, which explains why precise tuning of the rotation angles is crucial for balancing task objectives.

**Global Miss-Correction and Mean Task Loss.** Recall that the miss-correction angular error for task $i$ at iteration $t$ is $\varepsilon_i^t = \tilde{\alpha}_i^t - \alpha_i^*$. Define the deviation in task $i$'s loss due to miss-correction as

$$\Delta_i^t := L_i^t - L_i^*(\theta_t),$$

where $L_i^*(\theta_t)$ denotes the loss that would have been incurred using the optimal rotation angle $\alpha_i^*$.

As in Definition 2.8, we define the *global miss-correction angular error* at iteration $t$ as the average absolute angular deviation across tasks:

$$\text{GMAE}_t := \frac{1}{T} \sum_{i=1}^{T} |\varepsilon_i^t|.$$

Assuming the loss is locally smooth with respect to the rotation angle, we approximate the loss deviation using a second-order Taylor expansion:

$$\Delta_i^t \approx \frac{1}{2}(\varepsilon_i^t)^2 \cdot \left. \frac{\partial^2 L_i}{\partial \alpha_i^2} \right|_{\alpha_i = \alpha_i^*}.$$

Summing over all tasks gives a decomposition of the mean task loss:

$$\frac{1}{T} \sum_{i=1}^{T} L_i^t \approx \underbrace{\frac{1}{T} \sum_{i=1}^{T} L_i^*(\theta_t)}_{\text{Actual Gradient Effect}} + \underbrace{\frac{1}{2T} \sum_{i=1}^{T} (\varepsilon_i^t)^2 \cdot \left. \frac{\partial^2 L_i}{\partial \alpha_i^2} \right|_{\alpha_i = \alpha_i^*}}_{\text{Miss-Correction Deviation}}.$$

This shows that the miss-correction angular error directly contributes to the increase in average task loss. The more precisely each $\alpha_i$ approximates $\alpha_i^*$, the smaller the miss-correction deviation and the closer the overall learning process is to the optimal loss trajectory.

**Remark A.4.** *In practice, we cannot compute $\alpha_i^*$ exactly, but reducing the global miss-correction angular error* $\text{GMAE}_t$ *and its variance through alignment and task specificity promotes uniform convergence across tasks. This motivates the adaptive control of inner steps $\alpha_{\text{steps}}^t$ based on loss variability.*

### A.6 GLOBAL CONVERGENCE

We now establish the convergence of the RGB algorithm to a Pareto stationary point.

**Goal.** Using Theorem A.2 and Assumption A.1, prove that the RGB iterates converge to a Pareto stationary parameter vector.

**Theorem A.3** (Convergence to Pareto Stationarity). *Under Assumption A.1, the sequence $\{\theta_t\}_{t \geq 0}$ generated by Algorithm 1 converges to a Pareto stationary point $\theta^\star$.*

*Proof.* Define the Lyapunov (aggregate loss) function $\Phi(\theta) := \sum_{i=1}^{T} L_i(\theta)$. By $\beta$-smoothness of each $L_i$,

$$\Phi(\theta_{t+1}) \leq \Phi(\theta_t) - \eta_t \sum_{i=1}^{T} \langle \nabla L_i(\theta_t), v_t \rangle + \frac{\beta T}{2} \eta_t^2 \|v_t\|^2. \tag{$*$}$$

**Case 1 – $\theta_t$ is not Pareto stationary.** Theorem A.2 implies $\sum_i \langle \nabla L_i(\theta_t), v_t \rangle \geq c_t \|v_t\|$ with $c_t > 0$. For sufficiently small $\eta_t < c_t/(\beta T)$, the descent term dominates the quadratic error term in $(*)$, so $\Phi(\theta_{t+1}) < \Phi(\theta_t)$.

**Case 2 – $\theta_t$ is Pareto stationary.** Then by definition the rotation-adjusted direction vanishes ($v_t = \mathbf{0}$), so the right-hand side of $(*)$ equals $\Phi(\theta_t)$.

**Super-martingale argument.** Since $\Phi$ is lower-bounded (Assumption A.1(**A1**)) and the stepsizes satisfy the Robbins-Monro conditions (Assumption A.1(**A3**)), the Robbins–Siegmund almost-super-martingale lemma yields

$$\sum_{t=0}^{\infty} \eta_t \sum_{i=1}^{T} \langle \nabla L_i(\theta_t), v_t \rangle < \infty, \quad \text{and} \quad \lim_{t \to \infty} \|v_t\| = 0.$$

**Limit points are Pareto stationary.** Let $\theta^\infty$ be any accumulation point of $\{\theta_t\}$. Continuity of the gradients $\nabla L_i$ and of the rotation mapping $\theta \mapsto (\alpha^\star(\theta), v(\theta))$ implies $\|v(\theta^\infty)\| = 0$. Suppose $\theta^\infty$ were not Pareto stationary; then Theorem A.2 would give a *strictly* positive inner product $\langle \nabla L_i(\theta^\infty), v(\theta^\infty) \rangle > 0$, contradicting $v(\theta^\infty) = 0$. Therefore every limit point is Pareto stationary. If $\Phi$ is bounded or coercive, then the sequence $\{\theta_t\}$ cannot wander indefinitely. Since all its limit points are Pareto stationary, the entire sequence must converge to a single Pareto stationary point $\theta^\star$. □

## A.7 ABLATION STUDY: ADAPTIVE-STEP

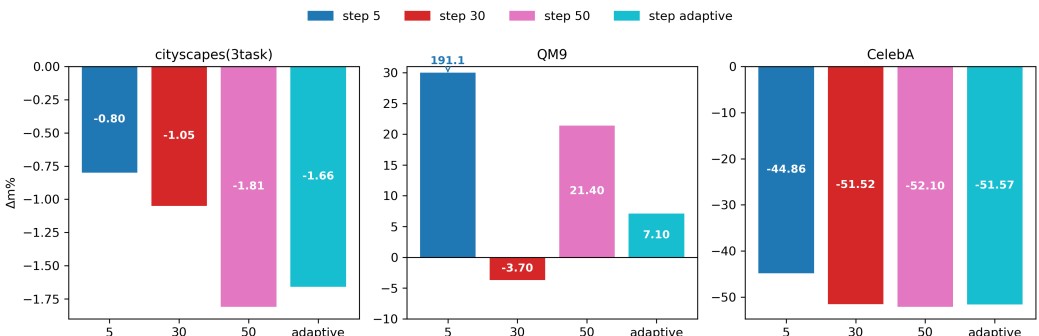

Figure 3: Comparison of performance across different step sizes (5, 30, 50, and adaptive) on Cityscapes, QM9, and CelebA.

**Discussion.** Figure 3 illustrates the impact of different step sizes across three datasets: QM9, Cityscapes, and CelebA. It can be observed that the optimal step size varies depending on the dataset. Specifically, QM9 achieves the best performance at **step 30**, and CelebA at **step 50**, while Cityscapes shows consistent improvements with larger step sizes, with the **adaptive step** variant yielding competitive results. The adaptive strategy proves particularly effective for datasets with more heterogeneous tasks, as it dynamically adjusts the exploration scale rather than relying on a fixed step size.

This behavior also appears to correlate with task complexity. For instance, CelebA, with 40 concurrent classification tasks, benefits from larger or adaptive step sizes that allow broader exploration of the optimization landscape. QM9, which contains moderately correlated regression tasks, achieves the best trade-off at step 30. Cityscapes, with three heterogeneous vision tasks, benefits from adaptive adjustment, achieving results comparable to fixed steps. This suggests that fixed steps may be suboptimal when task scales differ.

All ablation experiments were implemented using the publicly available FAMO repository as a base. Following its training protocol, we employed the STL configuration as the reference baseline and conducted all evaluations with a single run using **seed 0**, ensuring consistency across compared methods.

The mean loss curves observed in Figure 4 exhibit a clear relationship between training progression and the choice of step size ($\alpha_{\text{steps}}$). Early in the training process, all step sizes lead to similar improvements due to the presence of large gradient magnitudes, which dominate the early stages of optimization. However, as training progresses into the *converging regime*, differences between the step sizes become more apparent, and the curves begin to diverge. The divergence in the curves represents the varying effectiveness of different step sizes in the later stages of training. Models with well-chosen $\alpha_{\text{steps}}$ are better able to minimize loss, as they more accurately approximate the optimal rotation during training. This highlights the role of accurate rotations in reducing the global miss-correction angular error, which is crucial for improving convergence and task balancing.

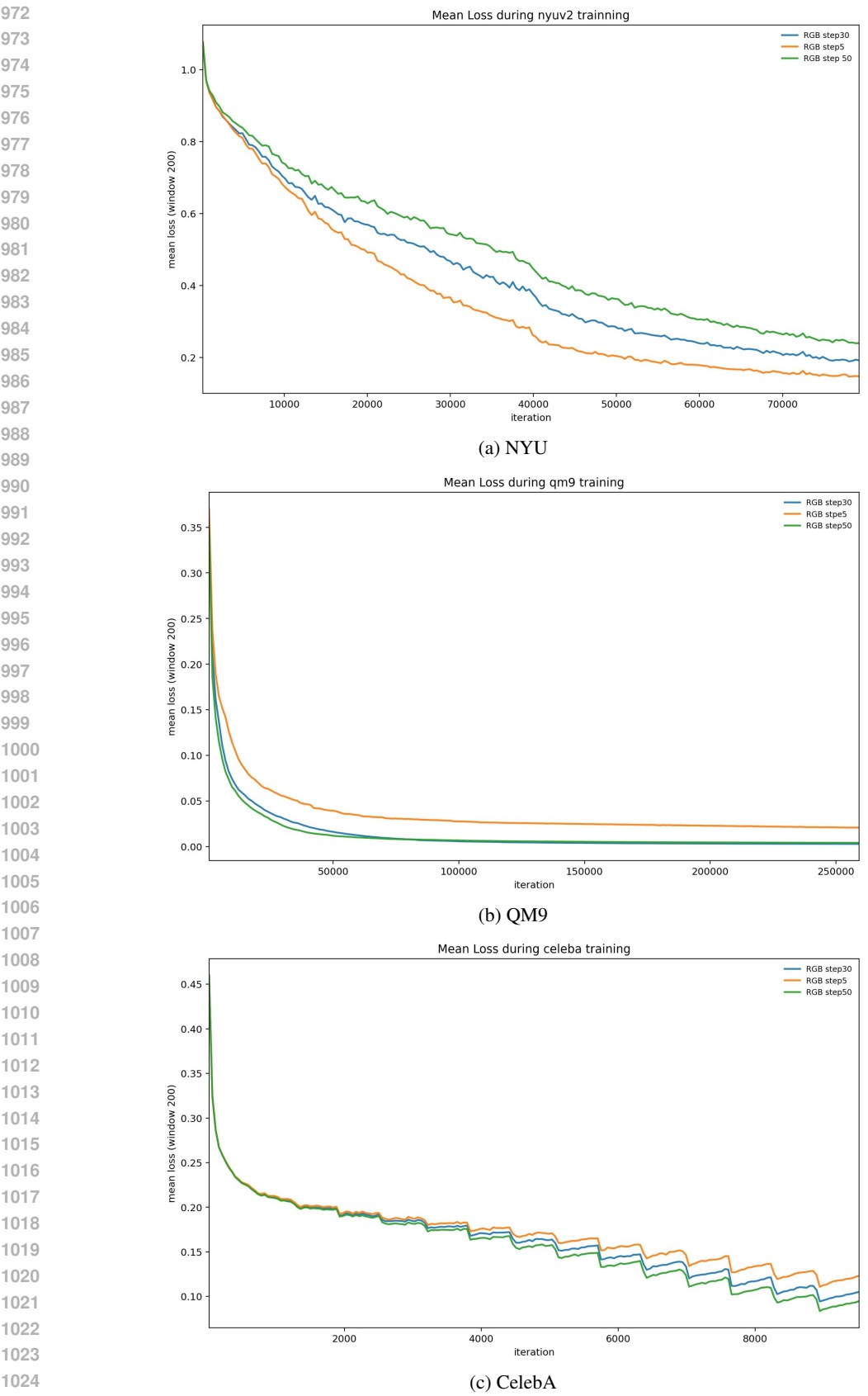

(a) NYU

(b) QM9

(c) CelebA

Figure 4: Mean loss curves across different $\alpha_{\text{steps}}$ values for NYU, QM9, and CelebA.

These observations underscore the difficulty of selecting a fixed, globally optimal step size $\alpha_{steps}$ across different datasets. Without prior knowledge of the task structure or the convergence behavior of the specific dataset, it is unclear which step size is ideal for any given domain. This creates an inherent challenge in hyperparameter tuning for gradient-based optimization, especially when transitioning across different datasets. To mitigate this issue, we propose an *adaptive-step* strategy that dynamically adjusts $\alpha_{steps}$ during training. Empirically, we find that this adaptive approach performs comparably to the original fixed-step RGB method, achieving competitive final mean losses across multiple datasets. The adaptive-step strategy provides clear advantages in terms of usability. By adjusting $\alpha_{steps}$ during training, it reduces the need for manual tuning of this hyperparameter, making the algorithm more robust and user-friendly. This is particularly useful in real-world applications where the exact characteristics of the data may not be known in advance.

Despite the practical benefits of the adaptive-step approach, it does not consistently recover the multi-task equilibrium relationships. Specifically, in the later stages of training, where gradient signals weaken, the adaptive method struggles to identify rotation angles that closely approximate the ideal optimal angles ($\alpha_i^*$). This limitation underscores a fundamental trade-off, while adaptivity simplifies hyperparameter tuning and enhances usability, it may not always guarantee the optimal correction of gradient misalignment. This observation emphasizes the balance between usability and performance, and suggests that future works could focus on further enhancing the adaptivity in these critical late stages of training.

## A.8 ANALYSIS OF MULTI-TASK EQUILIBRIUM RELATIONSHIP FOR GRADIENT DIRECT BALANCING METHOD

| NYU | Δm% (↓) | Mean Validation Loss (↓) | Conflict (↓) | Proximity (↓) | Condition Number (↓) | MER Balance (↑) |
|---|---|---|---|---|---|---|
| GradDrop | 5.27 | 0.115045369 | 0.444577545 | 0.119989514 | 9.00 | 0.248612289 |
| GradVac | 2.67 | 0.095970333 | 0.349338472 | 0.007405753 | 2.83 | 0.43826617 |
| PCGrad | 2.47 | 0.097575486 | 0.410914123 | 0.010193291 | 2.81 | 0.429126152 |
| RGB (Step-5) | -5.92 | 0.056488 | 0.2294147 | 0.0000221 | 1.60 | 0.559318683 |
| RGB (Step-30) | -3.03 | 0.067817986 | 0.4389925 | 6.33E-16 | 1.60 | 0.516992283 |
| RGB (Step-50) | -1.57 | 0.090802811 | 0.47516346 | 0.015465055 | 1.59 | 0.507688766 |

(a) NYU

| QM9 | Δm% (↓) | Mean Validation Loss (↓) | Conflict (↓) | Proximity (↓) | Condition Number (↓) | MER Balance (↑) |
|---|---|---|---|---|---|---|
| GradDrop | 438.00 | 0.012117501 | 0.451473355 | 0.124564335 | 37.27 | 0.12652434 |
| GradVac | 350.84 | 0.019525938 | 0.222009927 | 0.090152979 | 51.06 | 0.120078935 |
| PCGrad | 346.91 | 0.013420084 | 0.351806134 | 0.088938639 | 46.13 | 0.12005835 |
| RGB (Step-5) | 191.10 | 0.013297608 | 0.396466136 | 2.86E-10 | 25.75 | 0.163614919 |
| RGB (Step-30) | -3.70 | 0.001870396 | 0.386881411 | 0.00043226 | 27.71 | 0.158441847 |
| RGB (Step-50) | 21.40 | 0.002720715 | 0.431141198 | 4.60E-15 | 27.98 | 0.155277871 |

(b) QM9

| CelebA | Δm% (↓) | Mean Validation Loss (↓) | Conflict (↓) | Proximity (↓) | Condition Number (↓) | MER Balance (↑) |
|---|---|---|---|---|---|---|
| GradDrop | -46.55 | 0.136186212 | 0.338693649 | 0.137320668 | 37.72 | 0.130241761 |
| GradVac | -51.23 | 0.177525237 | 0.371290833 | 0.241720542 | 1007.09 | 0.024136463 |
| PCGrad | -48.85 | 0.115604043 | 0.376603216 | 0.178258911 | 176.69 | 0.058903859 |
| RGB (Step-5) | -44.86 | 0.103466474 | 0.504177034 | 2.41E-08 | 19 | 0.182320507 |
| RGB (Step-30) | -51.52 | 0.087983899 | 0.487084597 | 2.44E-07 | 18.73 | 0.184615673 |
| RGB (Step-50) | -52.10 | 0.076200992 | 0.504109144 | 1.81E-06 | 18.09 | 0.186619477 |

(c) CelebA

| Cityscapes | Δm% (↓) | Mean Validation Loss (↓) | Conflict (↓) | Proximity (↓) | Condition Number (↓) | MER Balance (↑) |
|---|---|---|---|---|---|---|
| GradDrop | 12.55 | 0.87046125 | 0.498722732 | 0.027725503 | 24.67 | 0.15903322 |
| PCGrad | -0.80 | 0.809684455 | 0.484813929 | 0.000199677 | 2.65 | 0.429510907 |
| RGB (Step-5) | -0.80 | 0.754312634 | 0.480502844 | 0.002237285 | 1.74 | 0.495946487 |
| RGB (Step-30) | -1.05 | 0.720389366 | 0.464729607 | 0.00815798 | 2.05 | 0.471201923 |
| RGB (Step-50) | -1.81 | 0.733408809 | 0.458433628 | 0.009760372 | 2.48 | 0.441731003 |

(d) Cityscapes

Figure 5: Analysis of Multi-task Equilibrium Relationship for Gradient Direct Balancing Method based on NYU, QM9, CelebA and Cityscapes

The statistics of Figure 5 comprises multiple indicators namely:

- $\Delta m\%$ is computed based on Eq 3.1.
- **Mean Validation Loss** refers to the minimum average loss across all tasks achieved during the training process. It represents the model's best convergent state and serves as the reference snapshot for the subsequent metrics.
- **Conflict** represents the average gradient conflict calculated specifically at the state of the *Mean Validation Loss*. It measures the degree of directional misalignment between task gradients at the optimal model checkpoint.

$$\text{Conflict} = \frac{1}{M} \sum_{(i,j)} \frac{1 - \langle g_i^{\text{adj}}, g_j^{\text{adj}} \rangle}{2}, \qquad M = \frac{T(T-1)}{2}. \tag{3}$$

- **Proximity** represents the average proximity term calculated specifically at the state of the *Mean Validation Loss*. It measures the squared distance between the rotated gradients and the original gradients, quantifying the preservation of task-specific information at the optimal checkpoint.

$$\text{Proximity} = \frac{1}{T} \sum_{i=1}^{T} \frac{\|r_i - \bar{g}_i\|^2}{4} \tag{4}$$

- **Condition Number** is computed from the eigenvalues of the PCA decomposition of the pairwise gradient conflict matrix. It implies the curvature of the optimization landscape and measures how anisotropic (uneven) the task-to-task conflict space is. A lower condition number indicates a smoother, more spherical optimization landscape. Formally:

$$\kappa = \frac{\lambda_{max}}{\lambda_{min}} \tag{5}$$

- **MER Balance** is a comprehensive geometric mean score used to examine and evaluate the state in accordance to Multi-task Equilibrium Relationship (MER). It aggregates the inverse scores of conflict, proximity, and the condition number. Higher values indicate a better equilibrium state. The formula is defined as:

$$\text{MER Balance} = \sqrt{\left(\frac{1}{1 + \text{Conflict}}\right) \cdot \left(\frac{1}{1 + \text{Proximity}}\right) \cdot \left(\frac{1}{1 + \text{Condition Number}}\right)} \tag{6}$$

**Discussion.** In this section, we attempt to analyze the impact of direct gradient manipulation on the global gradient system utilizing four diverse datasets, namely CelebA, QM9, NYUv2, and Cityscapes. Existing gradient-direct balancing methods, such as GradDrop Chen et al. (2020), GradVac Wang et al. (2020), and PCGrad Yu et al. (2020), serve as benchmarks to understand the effects of different operations on gradient properties.

To comprehensively evaluate these effects, we report the metrics at the state of minimum Mean Validation Loss, which represents the model's optimal convergent state. We assess the Conflict (directional misalignment) and Proximity (retention of task specific information) at this state. Furthermore, we introduce the Condition Number, computed from the PCA decomposition of the pairwise conflict matrix, to measure the anisotropy of the optimization landscape. It indicates the stability risks from misalignment and measures the perturbation sensitivity of the system. Thus, we aggregate these indicators into a MER Balance score (Eq. 6), where a higher value indicates a superior equilibrium between conflict reduction, information preservation, and optimization stability.

As shown in Figure 5, traditional methods often struggle to maintain equilibrium. For instance, on CelebA, PCGrad and GradVac achieve relatively low conflict scores (0.37 and 0.37) but exhibit significantly higher proximity scores (0.17 and 0.24) compared to RGB. This suggests a phenomenon

of over-correction 2.6, where these methods aggressively project gradients to reduce conflict, but in doing so, they distort the task-specific information (high proximity) and create a highly anisotropic optimization landscape. This is evidenced by their massive Condition Numbers on CelebA (176.69 for PCGrad and 1007.09 for GradVac), indicating an ill-conditioned solution space that hampers convergence, resulting in suboptimal $\Delta m\%$ performance (-48.85% and -51.23%). Similarly, Grad-Drop on QM9 exhibits a high Condition Number (37.27) and poor MER Balance, correlating with its inability to surpass the single-task baseline.

In contrast, RGB consistently achieves the highest MER Balance across all datasets. Specifically on CelebA, RGB (Step-50) maintains a remarkably low Proximity ($1.81 \times 10^{-6}$) and the lowest Condition Number (18.09). This indicates that RGB preserves the original task-specific information with high fidelity while successfully rotating the optimization landscape into a well-conditioned, isotropic state. Although RGB allows for a moderate level of raw conflict (0.50), the low Condition Number proves that this conflict is structured and manageable, rather than destructive. This balance of minimizing distortion and maximizing landscape smoothness is the essence of the Multi-Task Equilibrium Relationship (MER) 2.4. The empirical results confirm that this state of equilibrium directly correlates with superior generalization, as evidenced by RGB achieving the best $\Delta m\%$ of -52.10% on CelebA and -3.7% on QM9.

We specifically highlight the effectiveness of RGB on the QM9 dataset, where it achieves a $\Delta m\%$ of -3.7%, being the first method to surpass the average single-task baseline. This performance is directly attributed to RGB's ability to resolve high-dimensional conflicts without creating an ill-conditioned optimization landscape. As observed in the Figure 5, baselines like GradVac and PCGrad exhibit high condition numbers (51.06 and 46.13), indicating that their pairwise projections result in a highly uneven solution space when handling 11 conflicting tasks. In contrast, RGB maintains a significantly lower Condition Number (25.75). By optimizing rotation angles globally, RGB approximates a Multi-Task Equilibrium (MER) that balances conflict reduction with gradient fidelity, allowing it to navigate the complex loss landscape of QM9 more effectively than other pairwise projection methods. We note that while minimal steps (Step-5) can achieve high MER scores by strictly prioritizing Proximity, they may suffer from under-correction, which will be further discussed in Appendix A.9.

### A.9 ABLATION STUDY: OVER-CORRECTION AND UNDER-CORRECTION

**Analysis of Over-Correction and Under-Correction Based on Mean Loss.** As shown in Figure 4, the best performance for NYUv2 is achieved with $\alpha_{\text{steps}} = 5$, which corresponds to the lowest mean loss. This suggests that this step size closely approximates the optimal rotation, minimizing miss-correction. In contrast, $\alpha_{\text{steps}} = 50$ results in a higher mean loss, indicating over-correction of the gradients. A larger step size like 50 likely causes excessive adjustments to the rotation, misaligning the estimated and optimal rotation angles and hindering model learning.

For CelebA, the optimal performance is observed at $\alpha_{\text{steps}} = 50$. Here, the larger step size better facilitates convergence to the optimal rotation. On the other hand, a step size of 5 leads to under-correction, reflected in a higher mean loss. The small step size fails to sufficiently adjust the rotation, resulting in a larger angular error and slower convergence.

For QM9, the curves for $\alpha_{\text{steps}} = 30$ and $\alpha_{\text{steps}} = 50$ are nearly aligned, suggesting that the optimal step size lies somewhere between them. The $\alpha_{\text{steps}} = 5$ case again underperforms, confirming that smaller step sizes lead to under-correction. The results for $\alpha_{\text{steps}} = 30$ indicate that the ideal step size for this dataset is larger than 5 but may not need to be as large as 50.

**Analysis of Over-Correction and Under-Correction Based on Conflict and Proximity.** For NYUv2, RGB (Step-5) achieves the best overall $\Delta m\%$ and the lowest mean validation loss. Step-5 also has the lowest conflict and moderate proximity, indicating that the rotation is sufficient to reduce conflict while still preserving task-specific gradient directions. Increasing the step size to 30 and 50 causes the conflict to rise, which harms performance. Although Step-30 achieves the lowest proximity among the three variants, its higher conflict leads to worse performance than Step-5, re-

flecting a form of under-correction (insufficient alignment). Step-50 increases proximity relative to Step-5, but its conflict becomes even worse, indicating over-correction—the rotation overshoots and distorts the gradient geometry.

GradDrop exhibits both high conflict and high proximity, reflecting severe over-correction and poor MER Balance. GradVac and PCGrad reduce conflict more effectively than GradDrop, but still incur noticeably higher proximity than RGB, indicating that they continue to over-correct by moving gradients too far from their task-specific directions.

On QM9, RGB (Step-5) yields the worst $\Delta m\%$ and high conflict, despite having extremely low proximity. This is a classic example of under-correction: the small step size preserves task specificity but fails to sufficiently reduce conflict. RGB (Step-50) exhibits a similar pattern—low proximity but high conflict—also indicating under-correction. In contrast, RGB (Step-30) achieves a near-optimal balance between alignment and task specificity, with significantly lower conflict while keeping proximity small. This places Step-30 much closer to the MER-optimal balance for QM9.

GradDrop again displays the largest conflict and proximity, placing it firmly in the over-correction regime. GradVac and PCGrad reduce conflict more effectively than GradDrop, yet still suffer from high proximity and large condition numbers, indicating that they too over-correct the gradients.

For CelebA, RGB (Step-50) achieves the best $\Delta m\%$, the lowest mean validation loss, and the highest MER Balance. At Step-5, proximity is the lowest among all RGB variants, meaning gradients remain closest to their original forms. However, it also shows the highest conflict (0.487), indicating under-correction—the rotation is too small to properly align the task gradients. Increasing the step size to 30 and especially 50 reduces conflict while keeping proximity extremely small. Step-50 achieves the strongest balance between gradient alignment and preservation of task-specific information.

Among baselines, GradDrop shows both the smallest conflict and very large proximity, meaning gradients are aligned too aggressively while deviating far from their original directions—this is severe over-correction. GradVac and PCGrad reduce conflict moderately but maintain substantially larger proximity compared with RGB, again indicating over-correction, where gradients are forced to agree excessively at the cost of task-specific detail.

For Cityscapes, all RGB variants achieve similar mean validation losses, but RGB (Step-50) produces the best $\Delta m\%$ and a strong MER Balance.Step-5 has the lowest proximity and low conflict, indicating a very gentle correction. This corresponds to slight under-correction: the gradients are well preserved, but residual conflict is not fully resolved.Increasing the step size to 30 and 50 slightly raises proximity while keeping conflict and condition numbers low. Step-50 yields the best downstream performance, demonstrating that a somewhat larger rotation is beneficial and does not lead to harmful over-correction.

GradDrop exhibits both large conflict and large proximity, indicating extreme over-correction. PCGrad shows higher conflict but lower proximity than RGB, suggesting that it tends toward under-correction, as it does not sufficiently reduce conflict despite preserving task specificity better than GradDrop.

## A.10 The Convergence Analysis Based on Toy Experiment

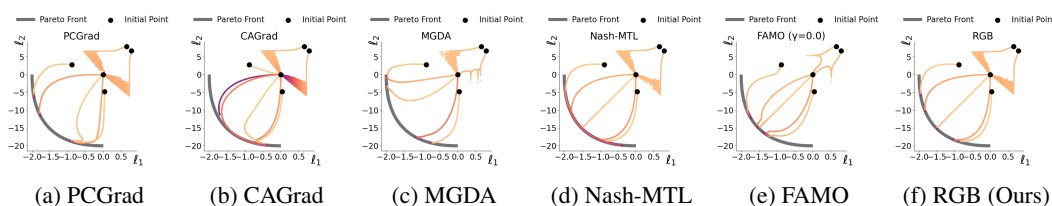

(a) PCGrad    (b) CAGrad    (c) MGDA    (d) Nash-MTL    (e) FAMO    (f) RGB (Ours)

Figure 6: Visualization of optimization trajectories on a toy 2-task problem. Each subplot corresponds to a different multi-task learning (MTL) method: PCGrad, CAGrad, MGDA, Nash-MTL, FAMO ($\gamma = 0.0$), and our proposed RGB. The axes denote the per-task losses $\ell_1$ and $\ell_2$, and the Pareto front is illustrated in dark gray. Compared to prior gradient manipulation methods, RGB drives the optimization closer to the Pareto front while maintaining stable convergence. All experiments were implemented using the publicly released FAMO repository and executed for 50000 steps under identical settings.

As shown, PCGrad and CAGrad exhibit similar tendencies: their trajectories fail to reach the Pareto front from all initialization points. MGDA improves upon this by driving all trajectories onto the front, but the converged points are scattered. In contrast, Nash-MTL and FAMO achieve smooth convergence with trajectories neatly landing on the front. Our proposed RGB method also converges cleanly to the front. Importantly, RGB is a *gradient-based* method, yet it achieves stable and consistent convergence comparable to weighting-based methods.

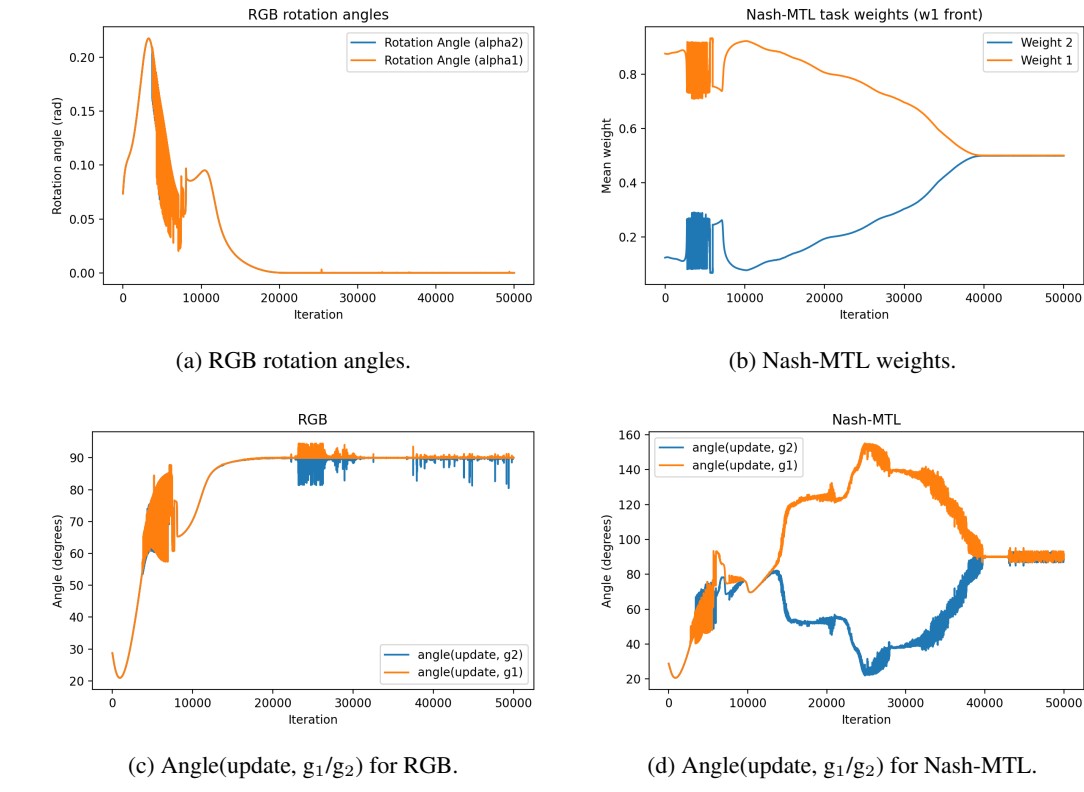

(a) RGB rotation angles.

(b) Nash-MTL weights.

(c) Angle(update, $g_1/g_2$) for RGB.

(d) Angle(update, $g_1/g_2$) for Nash-MTL.

Figure 7: Optimization dynamics in the toy FAMO experiment: rotation (RGB), reweighting (Nash-MTL), and update–gradient alignment.

The implementation of toy experiment is based on the FAMO repository. We would like to analysis the insight that the RGB and Nash-MTL exhibit markedly different internal dynamics but end up tracing very similar trajectories in the loss plane. Both methods have a similar objective of emphasizing the fairness of the update across tasks, but they achieve this in different ways. Nash-MTL is a weighting-based method that optimizes the task weights $w$ via the objective $G^\top G w = 1/w$, where $G \in \mathbb{R}^{d \times T}$ is the matrix of $T$ task gradients of dimension $d$. The update direction $d$ for Nash-MTL is then obtained as the weighted sum of task gradients based on the optimized $w$. For RGB, the rotation angles $\alpha_1$ and $\alpha_2$ quickly rise in the early iterations and then decay monotonically toward zero, indicating that the method performs moderate rotations only while the gradients are highly conflicting and then gradually switches off the correction once a stable compromise direction is found. Consistently, the angles between the RGB update and each task gradient converge to $\approx 90°$, so the shared update becomes nearly orthogonal to both individual gradients and thus aligned with a common descent direction close to a Pareto-stationary solution. Unlike Nash-MTL, RGB with a constant equal weight across the entire training phrase. Nash-MTL instead achieves compromise by reweighting: its task weights start strongly imbalanced, traverse a long transient phase, and finally converge nearly $0.5{:}0.5$ respectively, while the angles between the update and each task gradient oscillate widely and frequently cross $90°$, reflecting a bargaining-like process that alternately favors one task and then the other. Despite these different mechanisms (manipulation vs. reweighting), both methods enforce descent for all tasks and implicitly drive the inner products between the update and each gradient toward zero, so in this simple convex toy landscape they are attracted to essentially the same region of the Pareto front. This explains why the two-dimensional loss trajectories are visually similar—both sets of curves bend toward and then slide along the same Pareto frontier—while still revealing that RGB reaches this equilibrium with smaller angular fluctuations and a smoother, shorter path than Nash-MTL.

## A.11 ABLATION STUDY ON COEFFICIENTS $\mu$ AND $\lambda$

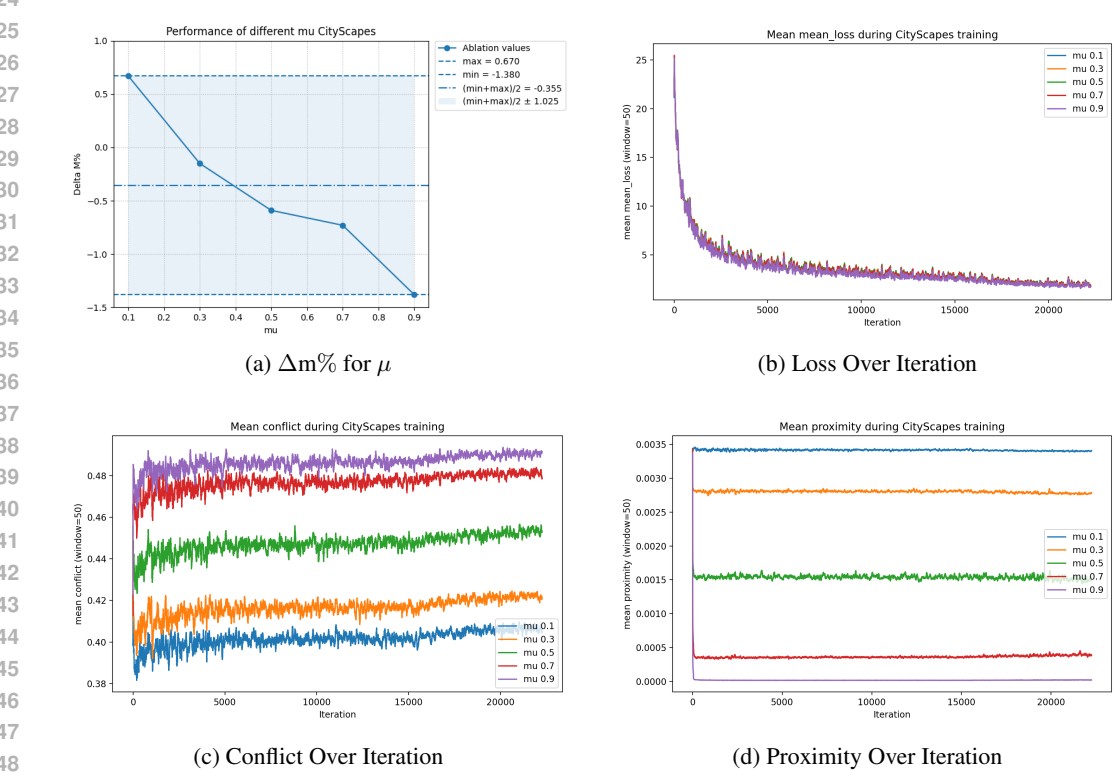

(a) $\Delta$m% for $\mu$

(b) Loss Over Iteration

(c) Conflict Over Iteration

(d) Proximity Over Iteration

Figure 8: Ablation Study of Coefficient $\mu$ on Cityscapes dataset.

We next analyze the effect of the EMA momentum parameter by sweeping $\mu \in \{0.1, 0.3, 0.5, 0.7, 0.9\}$. The $\Delta m\%$ curve in Figure 8 shows a clear monotonic trend: increasing $\mu$ steadily improves performance, with $\mu = 0.9$ achieving the best result. This reflects the importance of a slowly evolving global direction $d_t$ on this dataset. A larger momentum ensures that $d_t$ tracks a long-term consensus of the task gradients, rather than reacting strongly to the instantaneous noise of each training iteration.

The mean conflict trajectories show that smaller momentum values such as $\mu = 0.1$ and $0.3$ yield noticeably lower conflict. However, these configurations simultaneously produce the highest proximity values, indicating that the gradients are heavily distorted and pulled too far from their task-specific directions. This form of over-correction arises because a small $\mu$ causes $d_t$ to shift rapidly, leading the rotation operator to aggressively chase short-term fluctuations in the gradient field.

In contrast, increasing $\mu$ progressively reduces proximity, and by $\mu = 0.9$ the proximity values are lower than those at $\mu < 0.9$. The conflict at $\mu < 0.9$ is higher, but remains stable throughout training, suggesting that the larger momentum successfully prevents oscillatory over-adjustment of the gradients. The combination of minimal distortion (low proximity), moderate conflict, and smooth trajectories aligns well with the MER interpretation, and results in improved downstream task performance.

Overall, the $\mu$ ablation demonstrates that Cityscapes benefits from a high-momentum update rule, which stabilizes the global direction $d_t$ and prevents unnecessary rotation of the gradients. This observation reinforces our choice of fixing $\mu = 0.9$ as a universal setting across all datasets in our main experiments, as it consistently yields a favorable balance between task alignment and task-specific preservation.

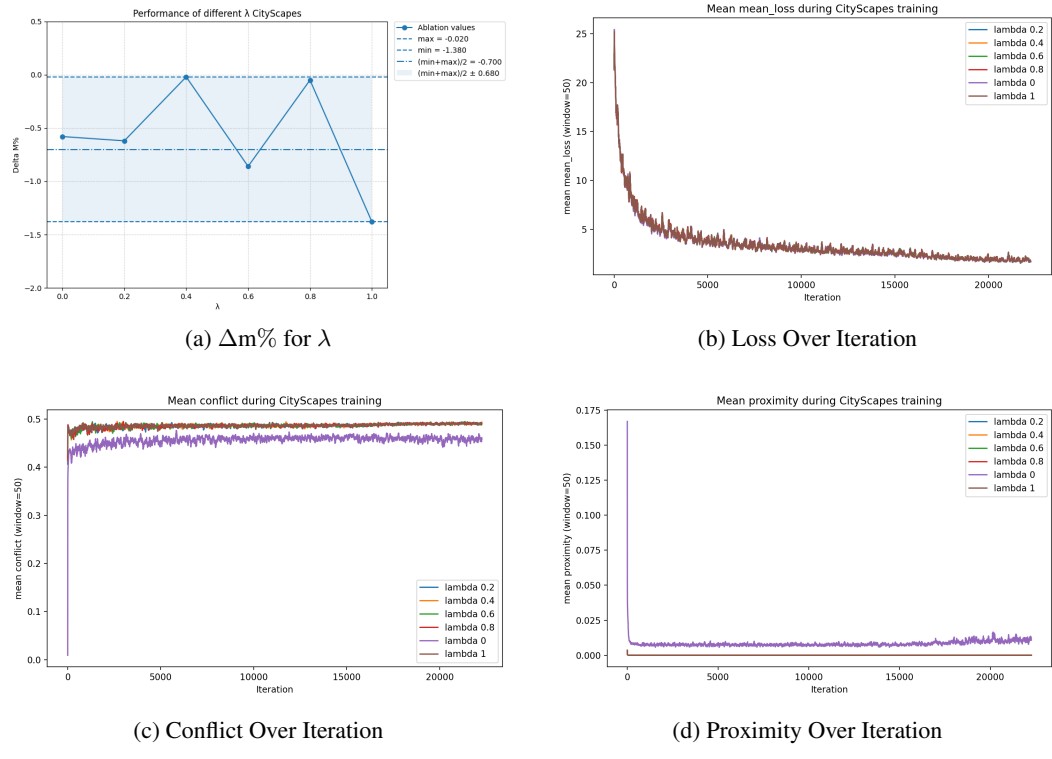

(a) $\Delta$m% for $\lambda$

(b) Loss Over Iteration

(c) Conflict Over Iteration

(d) Proximity Over Iteration

Figure 9: Ablation Study of Coefficient $\lambda$ on Cityscapes dataset.

We perform a sweep over $\lambda \in \{0, 0.2, 0.4, 0.6, 0.8, 1\}$ to examine how the RGB balance parameter shapes the optimization dynamics and downstream performance on the Cityscapes dataset. As shown in Figure 9, the overall validation performance is relatively flat across most choices of $\lambda$,

but the best $\Delta m\%$ is obtained at $\lambda = 1$. This indicates that the Cityscapes task structure favors the regime in which RGB strongly emphasizes the proximity term, thereby constraining the rotated gradients to remain closer to the original task directions.

The loss–iteration curves show that the mean loss decreases smoothly and consistently across all values of $\lambda$, indicating that the balancing coefficient do not disrupts the stability of convergence. Among the swept values, $\lambda = 0$ behaves noticeably differently from the other settings: it produces a slightly higher proximity at the start of training and a distinct conflict trajectory. This occurs because $\lambda = 0$ removes the proximity regularizer entirely, allowing the rotation to over-adjust toward conflict minimization without regard for preserving each task's gradient direction. As $\lambda$ increases, the proximity term gradually constrains this rotation, causing the curves for $\lambda = 0.2$ through $\lambda = 0.8$ to converge toward a similar geometric behavior.

This suggests that, for Cityscapes, excessive alignment of gradients across tasks is less helpful than preserving the task-specific descent directions. In other words, the dataset appears to reward solutions that maintain the semantic structure encoded in the original gradients, even at the cost of higher gradient disagreement.

## A.12 EXPERIMENT DETAIL

| Method | $\mu$ | $\alpha$ | $\epsilon_{\text{HOMO}}$ | $\epsilon_{\text{LUMO}}$ | $\langle R^2 \rangle$ | ZPVE | $U_0$ | $U$ | $H$ | $G$ | $C_v$ | $\Delta m\% \downarrow$ |
|---|---|---|---|---|---|---|---|---|---|---|---|---|
| | | | | | | MAE $\downarrow$ | | | | | | |
| RGB | 0.064 | 0.190 | 78.07 | 77.55 | 1.094 | 3.063 | 32.88 | 33.35 | 33.10 | 33.24 | 0.065 | -3.7 |
| $\pm$ stderr | $\pm$0.024 | $\pm$0.019 | $\pm$6.04 | $\pm$3.84 | $\pm$0.140 | $\pm$0.346 | $\pm$4.41 | $\pm$4.23 | $\pm$4.57 | $\pm$4.25 | $\pm$0.013 | $\pm$8.28 |

Table 4: Results on QM9. All metrics are MAE (lower is better). Results are averaged over four random seeds= 0,1,2,3.

| Method | Segmentation | | Depth | | Surface Normal | | | | | $\Delta m\% \downarrow$ |
|---|---|---|---|---|---|---|---|---|---|---|
| | mIoU $\uparrow$ | Pix Acc $\uparrow$ | Abs Err $\downarrow$ | Rel Err $\downarrow$ | Angle Dist. $\downarrow$ | | Within $t°$ $\uparrow$ | | | |
| | | | | | Mean | Median | 11.25 | 22.5 | 30 | |
| RGB | 41.93 | 67.56 | 0.529 | 0.224 | 24.74 | 19.30 | 29.11 | 56.72 | 69.43 | -5.92 |
| $\pm$ stderr | $\pm$0.42 | $\pm$0.28 | $\pm$0.013 | $\pm$0.0056 | $\pm$0.20 | $\pm$0.31 | $\pm$0.52 | $\pm$0.65 | $\pm$0.50 | $\pm$0.63 |

Table 5: Results on NYUv2. Hierarchical headers. Lower is better for $\downarrow$, higher for $\uparrow$. Our results are the average of three seeds = 0,1,2.

| Method | mIoU [%] $\uparrow$ | L1 [px] $\downarrow$ | MSE $\downarrow$ | $\Delta m\% \downarrow$ |
|---|---|---|---|---|
| RGB | 65.06 | 9.90 | 0.325 | -1.81 |
| $\pm$ stderr | $\pm$0.24 | $\pm$0.039 | $\pm$0.00046 | $\pm$0.17 |

Table 6: Results on Cityscapes (3 tasks). Lower is better for $\downarrow$, higher for $\uparrow$. Results are averaged over three random seeds=0,1,2.

| Method | Seed 0 ($\triangle$m%) | Seed 1 ($\triangle$m%) | Seed 2 ($\triangle$m%) | Average ($\triangle$m%) | MR |
|---|---|---|---|---|---|
| STL | 0 | 0 | 0 | 0 | 18.84 |
| LS | -50.29 | -13.16 | -20.24 | -27.90 | 10.06 |
| SI | -46.68 | -13.77 | -20.00 | -26.82 | 11.48 |
| RLW | -49.52 | -13.85 | -19.80 | -27.72 | 8.98 |
| DWA | -50.78 | -13.41 | -20.44 | -28.21 | 11.02 |
| UW | -50.79 | -13.79 | -17.94 | -27.51 | 10.02 |
| MGDA | -46.87 | -12.88 | -16.28 | -25.34 | 16.03 |
| PCGrad | -50.46 | -13.28 | -20.40 | -28.05 | 8.63 |
| GradDrop | -48.64 | -13.66 | -15.84 | -26.05 | 12.64 |
| CAGrad | -46.19 | -13.15 | -20.80 | -26.71 | 9.64 |
| IMTL-G | -50.14 | -14.02 | -19.96 | -28.04 | 8.61 |
| Nash-MTL | -50.29 | -14.00 | -20.76 | -28.35 | 8.58 |
| FAMO | -49.72 | -13.86 | -18.01 | -27.20 | 9.44 |
| GradVac | -51.23 | -13.72 | -20.81 | -28.59 | 7.91 |
| PIVRG | -51.45 | -13.10 | -17.63 | -27.39 | 9.28 |
| ConsMTL | -51.31 | -13.97 | -20.68 | -28.65 | **7.37** |
| RGB (Step-5) | -44.86 | -13.44 | -20.26 | -26.19 | 10.84 |
| RGB (Step-30) | -51.52 | -13.58 | -18.40 | -27.83 | 9.25 |
| RGB (Step-50) | -52.10 | -13.54 | -20.50 | **-28.71** | 9.26 |
| Adaptive RGB | -51.44 | -13.83 | -16.80 | -27.36 | 9.14 |

Table 7: Comparison Results of MTL Optimization Methods on CelebA, measured by $\triangle m$ and MR. Results are averaged over three random seeds=0,1,2.

**Analysis of Performance Variance Across Seeds.** As shown in Table 7, we observe significant variance in the absolute $\triangle$m% values across different random seeds while every variable remained constant expect random seeds. We attribute this volatility to three primary factors inherent to the 40 tasks CelebA benchmark. First, the joint optimization landscape for 40 concurrent tasks is highly non-convex and complex. Therefore, distinct random initializations (seeds) place the model in different basins of attraction, leading to convergence at disparate local minima. Second, the $\triangle m$% metric is relative to the Single-Task Learning (STL) baseline of that specific seed. Variations in the convergence quality of the 40 independent STL models per seed directly scale the denominator of the metric, amplifying fluctuations in the reported relative improvement. Third, balacing 40 tasks is inherently more unstable than balancing 3 (Cityscapes and NYUv2), where minor initial weight differences lead to vastly different optimization trajectories. Crucially, despite these shift in absolute magnitudes, the relative ranking of methods remains largely consistent, with RGB (Step-50) achieving the best average performance ($\triangle m$% = $-28.71$).

## A.13 COMPUTATIONAL COMPLEXITY AND EMPIRICAL OVERHEAD ANALYSIS

In this section, we provide a comprehensive theoretical and empirical analysis of time complexity, wall-clock training speed, and memory consumption to address the concerns regarding the computational overhead of the inner optimization loop and the scalability of Rotation-Based Gradient Balancing (RGB). All measurements reported in this section were conducted on the QM9 dataset (11 tasks), a standard benchmark for multi-task regression.

**Space Complexity.** RGB requires storing $T$ task gradients $\nabla L_i(\theta)$ and their corresponding orthogonal reference vectors $w_i$, resulting in a space complexity of $\mathcal{O}(TD)$, where $D$ is the number of shared parameters. This is consistent with standard gradient manipulation methods like PCGrad (Yu et al., 2020) and MGDA (Sener & Koltun, 2018), which also require storing individual task gradients.

**Time Complexity.** The primary computational cost arises from the optimization of Equation 1. The objective function involves calculating pairwise interactions between rotated gradients, scaling as $\mathcal{O}(T^2)$. With the inner loop running for $\alpha_{steps}$ iterations, the complexity added per training step is

$\mathcal{O}(\alpha_{\text{steps}} \cdot T^2 \cdot D)$. While this scales quadratically with tasks, the operations are performed on scalar angles or low-dimensional projections, making them computationally efficient on GPUs compared to the backward pass of the backbone network.

**Training Time and Backward Overhead.** We compare the wall-clock training time per epoch and the specific overhead of the backward pass (gradient computation + manipulation) in Figure 10a. As expected, simple weighting methods (RLW, DWA, UW) are the fastest as they involve only scalar multiplication. Among gradient manipulation methods, RGB demonstrates a computational footprint comparable to existing baselines. Specifically, RGB (Step-50) requires approximately **12.5 minutes per epoch**, which is equivalent to MGDA (12.5 min) and only marginally higher than PC-Grad (12 min). Looking closer at the backward pass overhead (Figure 10b), RGB (Step-50) incurs an overhead of **838.33 ms**, which is nearly identical to MGDA (837.3 ms). This confirms that even with the iterative inner loop, the vectorized angle optimization does not constitute a bottleneck for the 11-task system.

**Memory Usage.** Figure 10c illustrates the peak GPU memory usage. RGB consumes **3 GB** of peak memory, which is identical to PCGrad, MGDA, and Nash-MTL ( Navon et al. (2022)). The increase over simple weighting methods (2.9 GB) is negligible ($\approx 3\%$) and is solely due to the storage of individual task gradients before aggregation. This confirms that RGB introduces no significant memory overhead beyond standard gradient-based MTL requirements in this dataset.

**Scalability Conclusion.** While the global optimization of rotation angles introduces a training time complexity of $\mathcal{O}(T^2)$, this cost is essential for grounding and validating the Multi-Task Equilibrium Relationship (MER) theory. Our empirical results confirm that approximating MER through this global optimization significantly reduces global conflict while preserving task-specific features, which is a holistic balance that traditional pairwise methods fail to achieve. Furthermore, it is crucial to emphasize that the computational time and space complexity discussed herein exists solely during the Training Phase. In the Inference Phase (i.e., real-world application), the RGB model functions identically to standard single-task or multi-task models, utilizing only the optimized shared parameters $\theta$ for forward propagation. Consequently, RGB introduces zero additional latency during inference, ensuring it remains highly efficient for deployment. In summary, although there is a marginal increase in training cost, the MER-based rotation mechanism enables RGB to secure a more stable equilibrium among tasks. This effectively circumvents the oscillation and over-correction issues common in other gradient manipulation methods, ultimately yielding state-of-the-art (SOTA) performance.

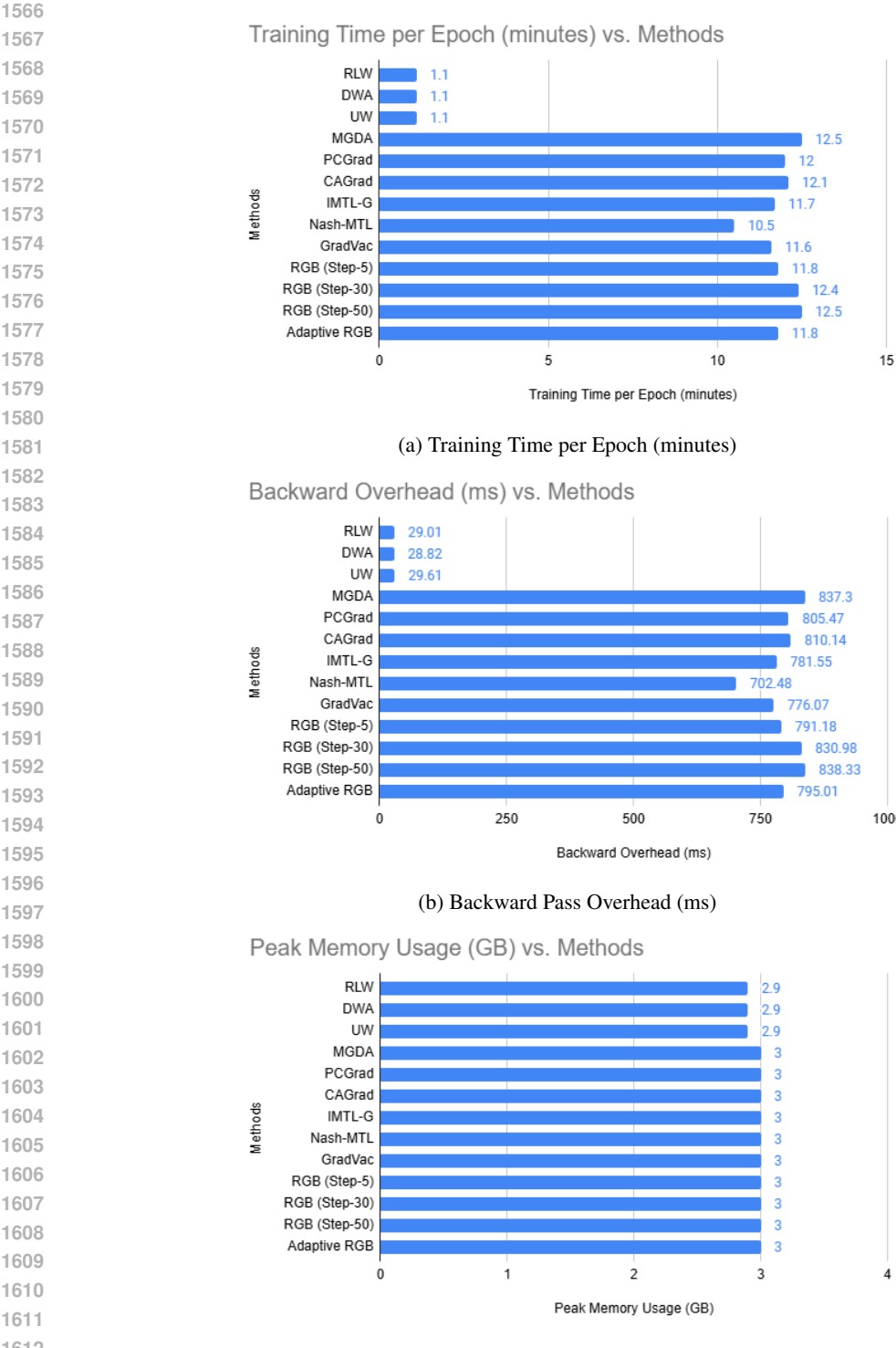

(a) Training Time per Epoch (minutes)

(b) Backward Pass Overhead (ms)

(c) Peak Memory Usage (GB)

Figure 10: Computational Overhead Comparison and Peak Memory Usage on QM9 (11 tasks).

### A.14 VISUAL ANALYSIS ON NYUv2 AND CITYSCAPES

To complement our quantitative results, we provide a visual comparison between our method (RGB) and a representative gradient manipulation baseline, PCGrad (Yu et al., 2020), on the NYUv2 and Cityscapes datasets. For NYUv2, we visualize predictions for three distinct tasks: Semantic Segmentation, Depth Estimation, and Surface Normal Prediction. For Cityscapes dataset, we visualize predictions for Semantic Segmentation and Disparity (Depth) Estimation.

**Semantic Segmentation on NYUv2 (Figure 11).** The segmentation results illustrate differences in intra-class consistency and boundary delineation. In the first row (kitchen scene), the PCGrad prediction for the cabinet exhibits noise and fragmented classifications. In contrast, RGB produces a more coherent region that aligns with the Ground Truth (GT). In the second row (bedroom scene), the RGB prediction retains thin structures, such as the bicycle frame and handlebars, which appear as disconnected segments in the PCGrad output.

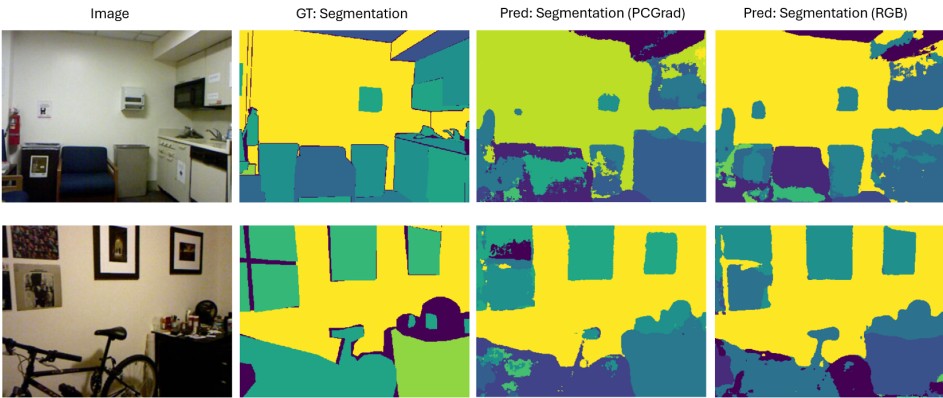

Figure 11: **Semantic Segmentation.** RGB (right) displays coherent intra-class regions on the cabinets (Row 1) and retains the structure of thin elements like the bicycle frame (Row 2) relative to PCGrad.

**Depth Estimation on NYUv2 (Figure 12).** The depth maps indicate differences in the preservation of fine structural details. In the second row, high-frequency details of the bicycle frame appear smoothed in the PCGrad prediction, whereas RGB retains the distinct silhouette of the bike against the wall. Additionally, in the first row, the PCGrad output exhibits artifacts on the back wall, while RGB produces a smoother and more accurate depth gradient across the room.

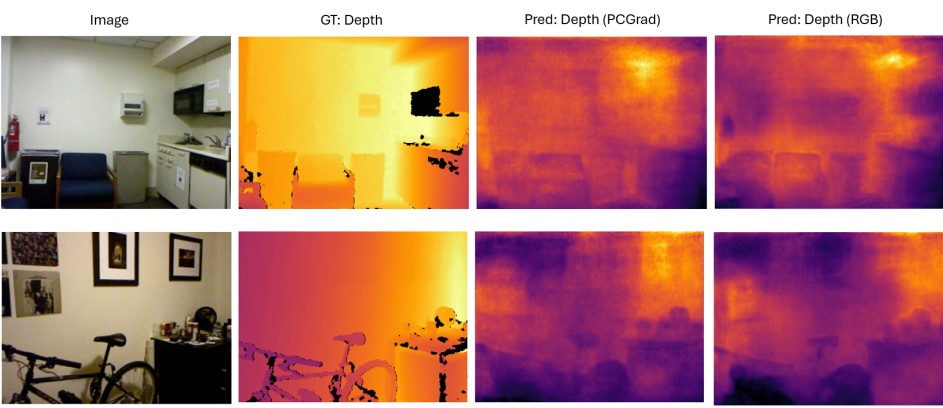

Figure 12: **Depth Estimation.** RGB retains high-frequency details, such as the bicycle structure (Row 2), whereas the PCGrad prediction appears smoother in these regions.

**Surface Normal Prediction on NYUv2 (Figure 13).** Surface normal visualization highlights the geometric consistency of the predicted planes. In the first row, the wall regions in the PCGrad prediction exhibit chromatic noise, indicating inconsistent variation in the predicted surface orientation. In contrast, RGB predicts uniform colors for planar surfaces (such as the walls and cabinets), suggesting a more consistent geometric representation. Similarly, in the second row, the RGB prediction clearly distinguishes the geometry of the bicycle and picture frames from the background wall, whereas the PCGrad output shows less distinct object boundaries.

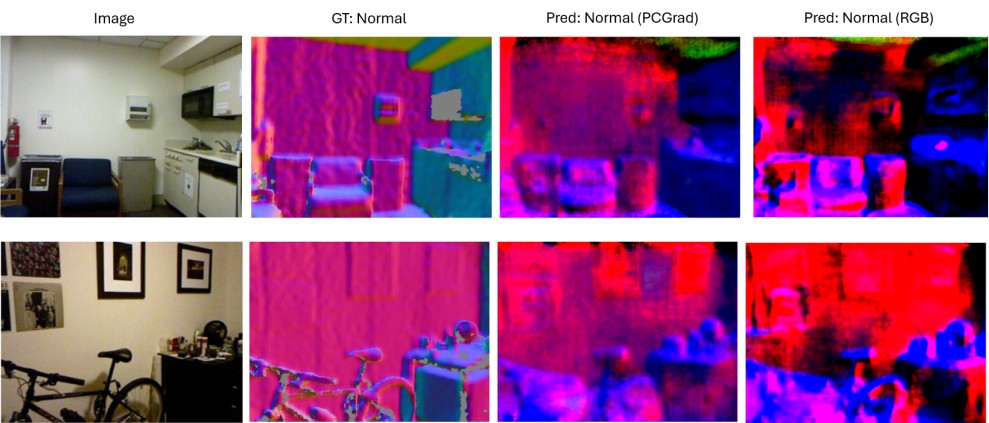

Figure 13: **Surface Normals.** PCGrad exhibits chromatic noise on planar surfaces (walls), whereas RGB produces more uniform predictions in these areas.

**Semantic Segmentation on Cityscapes (Figure 14).** The visualization highlights differences in how the methods handle fine-grained structures. In the first row, the traffic light poles and vertical background structures present a challenge. The PCGrad prediction (3rd column) exhibits fragmentation, where parts of the poles are disconnected or merged with the background. The RGB prediction (4th column) maintains improved continuity for these thin vertical elements, resulting in masks that are structurally closer to the Ground Truth (GT). In the second row, focusing on the distant pedestrians and vehicles, the PCGrad output tends to merge adjacent instances into larger, simplified regions. In contrast, the RGB output preserves more distinct boundaries between the small object instances (red or blue classes) and the background.

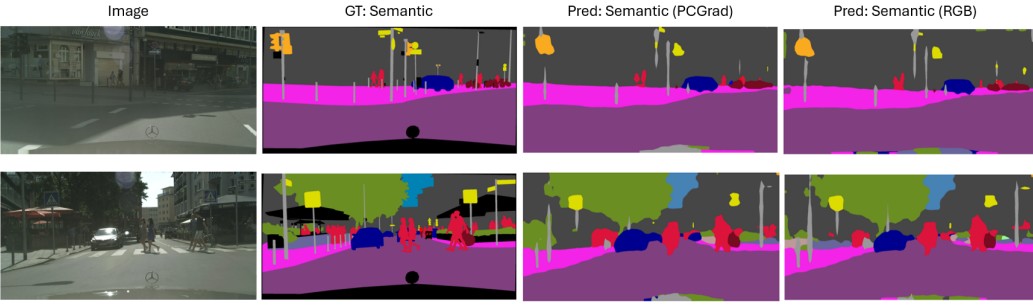

Figure 14: **Semantic Segmentation.** RGB (right) captures thin poles and small distant objects (pedestrians or cars) more accurately than PCGrad.

**Depth Estimation on Cityscapes (Figure 15).** The disparity maps reveal a distinct difference in edge preservation. In both rows, the PCGrad depth maps appear notably smoother, resulting in a blurred effect around object edges, which is particularly noticeable on the pedestrians in the second row. While this smoothing may reduce high-frequency noise, it also obscures object distinctness. In contrast, the RGB predictions retain sharper transitions between depths. This allows for clearer separation of foreground objects (such as the car hood and pedestrians) from the background environment, resulting in a visually sharper map with better definition compared to the smoothed output of PCGrad.

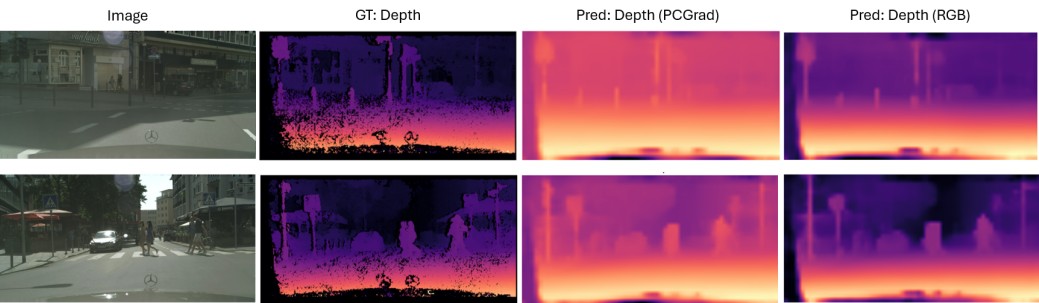

Figure 15: **Depth Estimation.** RGB produces sharper depth maps with better object delineation and fewer background artifacts compared to the blurry PCGrad output.

