# OpenReview forum: "Preserving Gradient Harmony: A Rotation-Based Gradient Balancing for Multi-Task Conflict Remedy"
_ICLR.cc/2026/Conference — Submitted to ICLR 2026_

### Official Review · Reviewer_vAwe · 2025-10-29

**Soundness:** 2
**Presentation:** 2
**Contribution:** 2
**Rating:** 2
**Confidence:** 5

**Summary:**

This paper proposes the Rotation-Based Gradient Balancing (RGB) algorithm, which rotates normalized task gradients toward a consensus direction by independently optimizing angle corrections for each task. The RGB algorithm offers fine-grained control that preserves beneficial gradient components, comprehensively reduces global gradient conflicts, and implicitly integrates loss change information to achieve balanced optimization. Empirical evaluations across four datasets show its effectiveness. However, there are still some aspects in this paper that require improvement or further clarification.

**Strengths:**

S1: The RGB algorithm offers precise control mechanisms that effectively preserve advantageous gradient components, systematically mitigate global gradient conflicts, and seamlessly integrate loss change information to facilitate balanced optimization.
S2: Empirical evaluations conducted across multiple datasets convincingly demonstrate the efficacy of the RGB algorithm.

**Weaknesses:**

W1: The innovation of the gradient rotation method is limited. Essentially, gradient rotation amounts to multiplying the original gradient by a rotated coordinate system. What is the fundamental difference between this approach and traditional gradient constraint or game-theoretic methods (such as CAGrad and Nash-MTL)?
W2: In practical training scenarios, gradients may be unstable or contain noise. How does the gradient rotation method address the issues of unstable gradients or noise?
W3: There is a lack of analysis on time and space complexity. A large number of gradient operations may pose challenges in terms of computational overhead when dealing with more tasks or larger models.
W4: The demonstrated effectiveness is not entirely robust. For instance, on the NYUv2 dataset, the gradient rotation algorithm only achieves state-of-the-art performance on the Segmentation task.
W5: There is a lack of visual analysis for visual tasks. For example, for the NYUv2 and Cityscapes datasets, it is recommended to provide visual results.
W6: There are doubts about the authenticity of the results. In Figure 6 of the appendix, why is the optimization trajectory plot of the Nash-MTL method labeled as "Ours"? Moreover, the optimization trajectory plots of the RGB method and the Nash-MTL method appear to be very similar. Are the presented results genuine?
W7: There are numerous formatting and layout issues. For example, the font size in Table 5 is too large, while the font sizes in Figures 4-5 are too small.

**Questions:**

Please see the weaknesses.

---

> ### Author Response · Authors · 2025-11-21
> **Response to Reviewer vAwe**
>
> ### W1: The innovation of the gradient rotation method is limited. Essentially, gradient rotation amounts to multiplying the original gradient by a rotated coordinate system. What is the fundamental difference between this approach and traditional gradient constraint or game-theoretic methods (such as CAGrad and Nash-MTL)?
>
> #### We understand your concern about the innovation of our paper, and we strive to highlight our novelty and contributions. Existing MTL balancing methods generally classified into gradient-manipulation (GM) and weighting-based (WB) method. As pointed by reviewer, traditional constraint-based (CAGrad) or game-theoretic (Nash-MTL) methods are WB methods, driven by first-order information to optimize tasks weight for directional aggregate, which is limited in capturing the gradient’s per-dimension sensitivities, potentially sub-optimize the convergence. Conversely, GM methods such as PCGrad, GradVac, GradDrop, etc. allow to directly manipulate the tasks gradient to avoid the directional update conflict.
>
> #### We hereby to clarify the reviewer question where the fundamental difference between RGB and traditional gradient constraint or game-theoretic methods (such as CAGrad and Nash-MTL) is the directness impact against the task gradient in which RGB allow to fine-gain manipulate the gradient to gradually reduce task conflict while highly retaining the task-specific information. In below, we would like to explain the uniqueness of RGB from GM methods.
>
> #### Unfortunately, we noticed that the existing GM methods are underperforming and overwhelmed by the weighting-based (WB) methods. Thus, we propose to reveal the insight of GM methods and point the current limitations of GM methods that hindering the development of this line of studies such as pairwise adjustment, operation’s order dependence, and projection rigor that omitting the global view of multi-task system, delivering inconsistencies performance.
>
> #### Given the current GM issue, we would like to claim our contribution and novelty where RGB is the first GM-based method’s paper to manipulate the task gradient in a global holistic view based on the global angular optimization via objective (1) that balance the global conflict resolve and retention of task-specific information, achieve the SOTA results.  We provide the detail insightful analysis for GM methods and propose the concept of multi-task equilibrium relationship that is supported by our updated empirical experiment A.8 across distinct datasets and inferring the phenomenon of miss-correction angular error A.9.
>
> ### W2: In practical training scenarios, gradients may be unstable or contain noise. How does the gradient rotation method address the issues of unstable gradients or noise?
>
> #### Thanks for your feedback. Regarding the unstable gradients issue, RGB mitigate the underlying impact at the beginning stage by introducing the EMA of normalized tasks gradients to smoothing the gradient noise and serving as reference direction $d_t$  for our angular objective optimization. Nature of EMA acting as a low-pass filter, smoothing out the single-batch noise or instability. Therefore, RGB is robust to unstable gradients precisely because its rotation target $d_t$ is a stable, smoothed and historical consensus. This is theoretically supported by Proposition A.2, which shows the convergence of the EMA update.
>
> ### W4: The demonstrated effectiveness is not entirely robust. For instance, on the NYUv2 dataset, the gradient rotation algorithm only achieves state-of-the-art performance on the Segmentation task.
>
> #### Thank you for your feedback. We hereby to provide a fine-gained result analysis to fairly evaluate the robustness of our method. We perform the percentage different analysis between the last SOTA result, ConsMTL (Qin et al., 2025b). Our RGB have a performance gain on segmentation tasks with two metrics respectively (+3.97%, +3.43%), but we fall short on depth estimation and surface normal prediction. For depth estimation two specific metrics with respective gain -3.59% in Abs Err, +4.00% in Rel Err. Lastly, we report the %different for every metrics of normal prediction (+1.60%, +2.66%, -6.31%., -2.69%, -1.27%). Nonetheless, RGB still remains explicitly competitive—often within 1.6%-6.31% relative performance to the current SOTA in Nyuv2 dataset, demonstrating consistent stability gains without catastrophic trade-offs for robustness testing.

---

> > ### Author Response · Authors · 2025-12-02
> >
> > For supplementary explanation for W1:
> >
> > #### The toy experiment (App. A.10, Fig. 7) further clarifies why RGB is fundamentally different from and can be preferable to weighting-based methods such as Nash-MTL: in this controlled setting, both methods reach essentially the same region of the Pareto front, so any observed differences stem purely from their optimization geometry rather than the final target. However, their internal dynamics are starkly different: Nash-MTL (WB) starts with highly imbalanced task weights that undergo a long bargaining-like transient, with angles between the update and each task gradient oscillating strongly and frequently crossing $90^\circ$, meaning the method keeps alternating which task is favored at each step; in contrast, RGB (GM, global) uses fixed equal weights, where rotation angles briefly increase only when gradients strongly conflict before decaying monotonically toward zero, and the angles between the RGB update and each task gradient converge smoothly to $\approx 90^\circ$ with small fluctuations, yielding a stable, symmetric compromise direction that preserves task-specific components. This experiment demonstrates that gradient rotation is not equivalent to reweighting in a different coordinate system—even when both converge to similar Pareto solutions, RGB achieves this via a globally optimized rotation that directly controls angular conflicts while retaining task information, whereas Nash-MTL relies on fluctuating task weights that induce oscillatory dynamics; in our large-scale experiments, this more stable and symmetric behavior of RGB translates into better final performance and easier hyperparameter tuning, supporting our claim that a globally optimized GM approach can surpass state-of-the-art WB methods.

---

> ### Author Response · Authors · 2025-11-21
> **Response to Reviewer vAwe**
>
> ### W6: There are doubts about the authenticity of the results. In Figure 6 of the appendix, why is the optimization trajectory plot of the Nash-MTL method labelled as "Ours"? Moreover, the optimization trajectory plots of the RGB method and the Nash-MTL method appear to be very similar. Are the presented results genuine?
>
> #### Thank for your feedback. Regarding the issue of the similar optimization trajectory result of our RGB method and Nash-MTL method, we have reproduced the result with our RGB code and confirmed the genuine of our results and not accidently incorrectly labelled “Nash-MTL” as ours. Both methods have a similar objective to emphasize the fairness of update but in different ways in which Nash-MTL as weighting-based method to optimize the tasks’ weight $w$ based on the objective function $G^T G w=1/w$, where $G$ represented a matrix of tasks’ numbers $t$ with dimension d of tasks’ gradients (d X t). Afterward the update direction d for Nash-MTL is the weighted sum of tasks’ gradient based on the optimized w.
>
> #### For our RGB as gradient-manipulation method, we aim to optimize the rotational angle a instead of tasks weight w, with the objective function in Equation 1 to manipulate the tasks gradient up to a global balancing state to minimize the global conflict, before combine into the update direction d. We have updated the additional convergence detail and analysis in appendix A.10 to show the difference between Nash-MTL and RGB.
>
> #### The prior gradient-manipulation based methods such as PCGrad exhibit unstable trajectories that “fail to reach the Pareto front”, revealing ineffective directional update due to gradient angular miss-correction issue, thus emphasizing our contribution in revealing the multi-task equilibrium relationship to harmonize the global multi-tasks system.
>
> ### W7: There are numerous formatting and layout issues. For example, the font size in Table 5 is too large, while the font sizes in Figures 4-5 are too small.
>
> #### Thank for your feedback. We have performed a thorough proofreading of the entire paper and appendix to fix all layout issues, especially Table 6, Figures 4 and 5.
>
> We have uploaded a first new version of the manuscript. Kindly review the latest revisions, which are highlighted in yellow, at your convenience. We will provide revisions for the remaining questions as soon as possible.

---

> ### Author Response · Authors · 2025-11-24
> **Response to Reviewer vAwe**
>
> ### W3: There is a lack of analysis on time and space complexity. A large number of gradient operations may pose challenges in terms of computational overhead when dealing with more tasks or larger models.
>
> #### Thank you for your feedback. We have added a detailed computational complexity analysis in Appendix A. 13, using the QM9 dataset (11 tasks) to empirically validate our method's efficiency. RGB has a space complexity of O(TD) (where D is the number of parameters and T is the number of tasks), which is required to store the individual task gradients. This is standard for gradient manipulation methods and consistent with baselines like PCGrad and MGDA. The primary cost is the inner optimization loop, which scales as O($\alpha_{\mathrm{step}} T^{2}D$) due to pairwise gradient interactions. Based on the empirical overhead results on QM9, RGB (Step-50) requires 12.5 minutes per epoch, which is equivalent to MGDA (12.5 min) and comparable to PCGrad (12 min). Besides that, RGB consumes 3 GB of peak memory, which is identical to PCGrad, MGDA, and Nash-MTL. The computational costs discussed above are exclusive to the training phase. For real-world deployment, RGB introduces zero additional latency during the inference phase, as the model functions identically to standard networks using the optimized shared parameters. In summary, although RGB incurs additional complexity, but this cost is justified by our method’s SOTA performance and able to approximate the MER.

---

> ### Author Response · Authors · 2025-11-27
> **Response to Reviewer vAwe**
>
> ### W5: There is a lack of visual analysis for visual tasks. For example, for the NYUv2 and Cityscapes datasets, it is recommended to provide visual results.
>
> #### Thank you for your suggestion. We have added the visual analysis in Appendix A. 14, comparing RGB against the representative gradient manipulation baseline, PCGrad on the NYUv2 and Cityscapes datasets.
>
> #### As illustrated in the figures for NYUv2, RGB produces masks with significantly better intra-class consistency for semantic segmentation, such as solid regions for cabinets, and sharper boundaries for thin structures like bicycle frames, compared to the fragmented predictions of PCGrad. In depth estimation, RGB preserves high-frequency details that are often blurred by PCGrad. For instance, the distinct silhouette of a bicycle against a wall is retained in RGB but washed out in the baseline. Furthermore, surface normal visualizations reveal that PCGrad exhibits noticeable chromatic noise on planar surfaces, indicating unstable orientation predictions, whereas RGB predicts smooth, uniform colors for these planes, reflecting a more geometrically consistent understanding of the scene.
>
> #### Similarly, for Cityscapes, RGB successfully captures fine-grained details in segmentation such as thin vertical traffic poles and small distant pedestrians, whereas PCGrad tends to miss or merge these instances. For depth estimation, RGB retains sharper transitions between objects at different depths, avoiding the over-smoothed, "foggy" artifacts observed in PCGrad depth maps that often obscure object distinctness.
>
> #### We have carefully considered each of the weaknesses raised and have incorporated updates to the manuscript to address them. We sincerely hope you will consider our responses and provide feedback in your final assessment.

---

### Official Review · Reviewer_xkvf · 2025-10-30

**Soundness:** 3
**Presentation:** 2
**Contribution:** 3
**Rating:** 6
**Confidence:** 4

**Summary:**

This paper investigates multi-task learning (MTL) from the perspective of gradient manipulation. It argues that previous approaches often tend to over- or under-correct task gradients, thereby compromising task-specific gradient information. To address this issue, the paper proposes a rotation-based strategy that introduces an objective function designed to simultaneously minimize gradient conflicts and regularize gradient deviations. Extensive experiments conducted on four public datasets demonstrate the competitive performance of the proposed method.

**Strengths:**

1. The idea of de-conflicting task gradients through a rotation-based perspective appears to be novel in the context of MTL.

2. The proposed method achieves competitive performance across multiple mainstream MTL benchmarks.

3. The paper provides some theoretical insights that help support and motivate the proposed approach.

**Weaknesses:**

1. Figure 1 provides a conceptual illustration of a phenomenon. To move beyond a purely illustrative claim, the manuscript would be significantly strengthened by either empirical evidence or a formal theoretical analysis to verify and substantiate this depiction.
2. In Figure 1, how to derive such an optimal gradient direction?
3. Can you provide some insights on why RGB is extremely effective on QM9?
4. Why only report across a single seed on CelebA? And why is $\Delta m$ \% here different from previously reported in other literature [1]?
5. The notations $f_z(x)$ and $F(x)$ are employed without prior definition.

Reference:

[1] Fair resource allocation in multi-task learning. ICML 2024.

**Questions:**

Please refer to the Weaknesses.

---

> ### Author Response · Authors · 2025-11-21
> **Response to Reviewer xkvf**
>
> ### W2: In Figure 1, how to derive such an optimal gradient direction?
>
> #### Thank for your feedback. We would like to clarify that the “Optimal Gradient” in Figure 1 is a theoretical construct for illustration. Instead, it represents the ideal gradient state in accordance with our definition of Multi-task Equilibrium Relationship (MER), which we introduce in Definition 2.4. RGB is designed to approximate the MER state by optimizing Eq1 for the optimal rotation angles ${α_i^* }_{(i=1)}^T$  to balance the global multi-tasks alignment and proximity (retention of task specific information). The existence of this optimal solution is guaranteed by Theorem A.1. The “Adjusted Gradient” is the result of different gradient manipulation methods, and we define the gap between MER and adjusted gradient as Miss-Correction Angular Error (MCAE). We have updated manuscript for the experiment detail of different datasets such as NYUv2, QM9, Cityscapes and CelebA in Appendix A.8 to have a stronger empirical inference for MER and MCAE in A.8 and A.9.
>
> ### W5: The notations $f_z (x)$ and $F(x)$ are employed without prior definition.
>
> #### Thank for your feedback. We have corrected the notations in Section 2.1 to aligned them with Section 2.2. Specifically, we added a statement clarifying that $x$ represents the model parameters (equivalent to $θ$) and $f_z (x)$ represents the loss for task $z$ (equivalent to $L_i (θ)$), where $Z$ denotes the total number of tasks (equivalent to $T$). This ensures consistent notation throughout the paper.
>
> We have uploaded a first new version of the manuscript. Kindly review the latest revisions, which are highlighted in yellow, at your convenience. We will provide revisions for the remaining questions as soon as possible.

---

> ### Author Response · Authors · 2025-11-22
> **Response to Reviewer xkvf**
>
> ### W1: Figure 1 provides a conceptual illustration of a phenomenon. To move beyond a purely illustrative claim, the manuscript would be significantly strengthened by either empirical evidence or a formal theoretical analysis to verify and substantiate this depiction.
>
> #### Thank you for your suggestion. We would like to refer to Appendix A.8 and A.9, which provide the empirical evidence by quantitatively verifying the phenomena of Multi-task Equilibrium Relationship (MER) and miss-correction based on the indicator of conflict assessment, proximity measurement, and MER Balance metrics. Specifically, we analyse the mean loss curves in Figure 4. We show that for CelebA, $α_{steps}=5$ results in a worse mean loss curve, as the insufficient optimization steps (which we define as under-correction) result in a large angular error, preventing the gradients from reaching the optimal Multi-Task Equilibrium. Conversely, for NYUv2,$α_{steps}=50$ leads to a higher mean loss, as the excessive rotation (an over-correction) causes significant deviation from the original task-specific gradients.
>
> #### We further support this in Figure 5, where other methods like PCGrad and GradVac show signs of over-correction by having high proximity scores (deviating too far from the original gradient). These baselines also have massive Condition Numbers and lower MER Balance scores compared to RGB. We have added a forward reference in Figure 1's caption to explicitly link this conceptual illustration to the supporting empirical evidence in the appendix.
>
> ### W3: Can you provide some insights on why RGB is extremely effective on QM9?
>
> #### Thank for your feedback. We would like to reveal the reason in which RGB to excel on the 11-tasks QM9 benchmark. RGB is effectively resolving the high-dimensional conflicts while preventing the ill-conditioned landscapes often created by pairwise projection methods (such as PCGrad) as analyzed in Appendix A.8. Our results show that RGB maintains a significantly lower Condition Number, ensuring a smoother optimization trajectory that allows it to successfully approximate the Multi-Task Equilibrium Relationship (MER) in this high-dimensional setting. Consequently, RGB achieves the highest MER Balance score, which fundamentally balances the trade-off between minimizing global conflict and preserving gradient fidelity. Unlike PCGrad and GradDrop, which rely on aggressive gradient projection, RGB optimizes precise rotations to resolve conflicts while maintaining the original task information (proximity), allowing it to effectively surpass the single-task baseline.
>
> We have updated Appendix A.8 to include additional datasets that further demonstrate the robustness of our method. We have also added extra indicators, such as the MER Balance Score and the Condition Number, to provide a more comprehensive analysis. Kindly review the latest revisions, which are highlighted in yellow, at your convenience. We will address the remaining questions as soon as possible.

---

> > ### Comment · Reviewer_xkvf · 2025-11-27
> >
> > Thank you for your efforts in addressing my concerns. I am still concerned about the fourth weakness; I believe you can conduct an appropriate evaluation according to FAMO's official implementation [1].
> >
> > Reference:
> >
> > [1] FAMO: Fast Adaptive Multitask Optimization. NeurIPS 2023.

---

> > > ### Author Response · Authors · 2025-11-28
> > > **Replying to Official Comment by Reviewer xkvf**
> > >
> > > #### Thank you for your response. We are currently conducting the evaluation on CelebA using the official FAMO implementation across multiple seeds, as requested. We are actively generating these results and will update the paper and this discussion thread with the new table as soon as the experiments conclude.
> > >
> > > #### Regarding the difference in reported  $\Delta m$% compared to other literature, to ensure a fair comparison, we re-implemented all methods within a single, unified codebase (FAMO). Specifically, as noted in the paper, existing works did not release the specific baseline results required for this comparison. Therefore, we re-ran the Single Task Learning (STL) configuration ourselves based on the FAMO repository to serve as the reference baseline. The reported $\Delta m$% is calculated against this re-run STL baseline.

---

> > > > ### Author Response · Authors · 2025-12-02
> > > >
> > > > Thank you for your overall suggestions. We acknowledge that our initial submission reported results for a single seed (Seed 0) due to the significant computational cost of the 40-task benchmark. To address this concern and validate robustness, we have now re-evaluated all methods across three random seeds (0, 1, 2) using the official FAMO implementation. The average of 3 seed's result as presented in Appendix A.12, Table 7, maintained our last conclusive performance to have the best average performance (Δm%=-28.71), confirming the robustness of our method across different initializations.

---

### Official Review · Reviewer_nN2S · 2025-10-31

**Soundness:** 3
**Presentation:** 3
**Contribution:** 3
**Rating:** 4
**Confidence:** 3

**Summary:**

RGB is a rotation-based gradient balancing method for multi-task learning: each task’s unit gradient is rotated toward an EMA reference via a global alignment-plus-proximity objective, then averaged. The paper proves convergence to Pareto-stationary points and reports consistent gains on NYUv2, Cityscapes, CelebA and QM9.

**Strengths:**

1. Principled, globally coordinated reconciliation. Uses per-task scalar rotations toward a shared reference to reduce conflict globally, avoiding PCGrad-style pairwise projections while preserving task-specific signal via a proximity term.

2.Theoretical grounding to Pareto stationarity. Provides existence of the inner minimizer and convergence to Pareto-stationary solutions under standard smoothness/stepsize assumptions, tying the geometric construction to multi-objective optimality rather than a heuristic.

**Weaknesses:**

1.Mathematical clarity in the rotation operator.
The text states that $r_i(\tilde\alpha_i)=d_t$ when $\tilde\alpha_i=\pi/2$, but by construction $r_i(\pi/2)=w_i$ (the component orthogonal to $\bar g_i$). Hitting $d_t$ requires $\tilde\alpha_i=\angle(\bar g_i,d_t)$.

2.Unquantified compute/memory overhead and scalability.
The method introduces an EMA reference direction and an inner loop for angle optimization, but the paper does not report peak memory, extra FLOPs, or wall-clock overhead vs. baselines, nor scaling with the number of tasks $T$.

3.Missing ablation on the proximity weight $\lambda$.
$\lambda$ governs the trade-off between conflict reduction and fidelity to original gradients, effectively bounding the rotation. There is no systematic sweep . A grid over $\lambda$ with curves for scores is needed to validate robustness and to show the method does not hinge on a narrow setting.

4.Missing ablation on the EMA coefficient $\mu$ and reference-direction design.
Because $d_t$ defines the feasible rotation plane, its construction is critical to stability. The paper lacks sensitivity analyses over $\mu$ and comparisons to alternatives.

5.Incomplete positioning relative to prior rotation methods (RotoGrad)[1].
The paper does not cite or empirically compare to RotoGrad, a closely related line that also uses rotation to harmonize multi-task gradients. A proper literature positioning and a head-to-head comparison under the same backbone/budget are needed.

[1] Javaloy, A. & Valera, I. RotoGrad: Gradient Homogenization in Multitask Learning. ICLR 2022.

**Questions:**

See the weaknesses part.

---

> ### Author Response · Authors · 2025-11-21
> **Response to Reviewer nN2S**
>
> ### W1: Mathematical clarity in the rotation operator. The text states that $r_i (α ̃_i )=d_t$ when $α ̃_i=π/2$, but by construction $r_i (π/2)=w_i$ (the component orthogonal to $g ̅_i$). Hitting $d_t$ requires $α ̃_i=∠(g ̅_i,d_t)$.
>
> #### Thank you for your feedback. We acknowledge that the statement on Page 5 claiming that $r_i (π/2)$ hits the reference direction $d_t$ was imprecise. As correctly noted by the reviewer (and as defined in our Appendix Proposition A.1), our construction yields $r_i (π/2)=w_i$. To clarify, $w_i$  serves as the geometric reference that defines the solution space for the rotation by specifically selecting the orthogonal vector nearest to $d_t$. This ensures that the gradient rotates toward the consensus direction $d_t$ (within the plane defined by $g ̅_i$ and $w_i$., rather than towards an arbitrary orthogonal direction. Therefore, our operator interpolates between the gradient and this orthogonal projection, effectively moving toward the consensus. We have revised the text on Page 5 to correctly state that the rotation interpolates between $g ̅_i$ and $w_i$ (the orthogonal component derived from $d_t$), effectively rotating the gradient toward $d_t$.
>
> ### W4: Missing ablation on the EMA coefficient  μand reference-direction design. Because $d_t$ defines the feasible rotation plane, its construction is critical to stability. The paper lacks sensitivity analyses over $μ$ and comparisons to alternatives.
>
> #### Thank for your suggestion to comprehend the studies in which we agree that the EMA coefficient $μ$ is relatively important in determining the reference-direction design. Based on this, we construct the ablation studies to examine the hyperparameter $μ$ in a range of {0.1, 0.3, 0.5, 0.7, 0.9} to perform sensitivity analysis. We have additionally included the experiment result and its detail analysis in appendix A.11.
>
> We have uploaded a first new version of the manuscript. Kindly review the latest revisions, which are highlighted in yellow, at your convenience. We will provide revisions for the remaining questions as soon as possible.

---

> ### Author Response · Authors · 2025-11-24
> **Response to Reviewer nN2S**
>
> ### W2: Unquantified compute/memory overhead and scalability. The method introduces an EMA reference direction and an inner loop for angle optimization, but the paper does not report peak memory, extra FLOPs, or wall-clock overhead vs. baselines, nor scaling with the number of tasks $T$.
>
> #### Thank you for your feedback. We have added a detailed computational complexity analysis in Appendix A. 13, using the QM9 dataset (11 tasks) to empirically validate our method's efficiency. RGB has a space complexity of O(TD) (where D is the number of parameters and T is the number of tasks), which is required to store the individual task gradients. This is standard for gradient manipulation methods and consistent with baselines like PCGrad and MGDA. The primary cost is the inner optimization loop, which scales as O($\alpha_{\mathrm{step}} T^{2}$D) due to pairwise gradient interactions. Based on the empirical overhead results on QM9, RGB (Step-50) requires 12.5 minutes per epoch, which is equivalent to MGDA (12.5 min) and comparable to PCGrad (12 min). Besides that, RGB consumes 3 GB of peak memory, which is identical to PCGrad, MGDA, and Nash-MTL. The computational costs discussed above are exclusive to the training phase. For real-world deployment, RGB introduces zero additional latency during the inference phase, as the model functions identically to standard networks using the optimized shared parameters. In summary, although RGB incurs additional complexity, but this cost is justified by our method’s SOTA performance and able to approximate the MER.
>
> ### W3.Missing ablation on the proximity weight governs the trade-off between conflict reduction and fidelity to original gradients, effectively bounding the rotation. There is no systematic sweep. A grid over with curves for scores is needed to validate robustness and to show the method does not hinge on a narrow setting.
>
> #### We appreciate your feedback to improve the study in which we agree that the λ coefficient is important to retain task-specific information. Based on this, we construct the ablation studies to examine the hyperparameter λ in a range of {0, 0.2, 0.4, 0.6, 0.8, 1} to perform a robust analysis. We have additionally included the experiment result and its detail analysis in appendix A.11. In short, the ablation on the balancing coefficient λ produces a more “wavy” Δm% curve: performance goes up and down with different λ, so there is no single value that is uniformly optimal in all settings. However, the magnitude of these variations is moderate, and all choices of λ remain competitive, with overall performance consistently below the STL baseline. This suggests that the method is quite robust to λ: changing λ mainly induces mild fluctuations around a strong baseline rather than causing severe degradation.

---

> ### Author Response · Authors · 2025-11-27
> **Response to Reviewer nN2S**
>
> ### W5: Incomplete positioning relative to prior rotation methods (RotoGrad)[1]. The paper does not cite or empirically compare to RotoGrad, a closely related line that also uses rotation to harmonize multi-task gradients. A proper literature positioning and a head-to-head comparison under the same backbone/budget are needed.
>
> #### Thank you for your insightful feedback. We appreciate your suggestion to include a head-to-head comparison with RotoGrad, and we have made the necessary updates in the manuscript. First, RotoGrad attempts to align all task gradients for a consensus direction to mitigate the directional conflict by rotating task gradients via learning-based rotation parameters. However, RotoGrad might neglect task-specific information which causing a deviation from the optimal update. For gradient manipulation methods, it is essential to consider the question of how much task-specific information should be sacrificed to maintain a consensus directional update. The fundamental difference is that our RGB aims to reduce gradient conflict between tasks while emphasizing the preservation of task-specific information. RGB realizes the goal above via a rotation operator, where the rotation angle $\alpha$ is learned via the combined objective between global task alignment (pairwise misalignment) and gradient proximity (distance between original and adjusted gradients). In our paper, we propose the concept of multi-task equilbirum relationship (MER), in which the ideal state to balance between the perservation of task-specific information and global alignment. We have provided the emperical support for the MER state in distinct datasets with the insight behind.
>
> #### In terms of complexity, RotoGrad introduces significant overhead, as the rotation matrices are learned during training and are updated per iteration. The overall time complexity of RotoGrad is $O(KD^3)$, which increases with both the number of tasks and the feature dimension. This leads to higher computational costs for both training and inference due to the per-iteration updates and the need to store and update task-specific rotation parameters. RGB operates in gradient space and introduces no additional model parameters. Its time complexity is $O(\alpha_{steps} T^2 D)$ and does not incur any inference overhead. The method preserves task-specific identities while finding a common direction, leading to a smaller computational cost in terms of both memory and training time. We have included the difference between RGB and RotoGrad in the introduction section of the updated manuscript.
>
> #### We have carefully considered each of the weaknesses raised and have incorporated updates to the manuscript to address them. We sincerely hope you will consider our responses and provide feedback in your final assessment.
>
> #### [1] Javaloy, A. & Valera, I. RotoGrad: Gradient Homogenization in Multitask Learning. ICLR 2022.

---

### Author Response · Authors · 2025-11-26
**Looking Forward to Reviewers’ Feedback on Our Response**

We sincerely appreciate the reviewers’ insightful comments. We sincerely hope that the reviewers will consider our responses and provide their feedback.

---

> ### Author Response · Authors · 2025-12-02
>
> We strive our best to address the reviewer concern about our study and would like to provide a brief summary of how we have addressed the concerns raised by three reviewers to assist in your evaluation. We have engaged extensively with the reviewers and implemented significant updates to the manuscript and appendix.
>
> ## Response to Reviewer nN2S
>
> The response focused on improving mathematical clarity and distinguishing the method from prior work. We refined the definition of the rotation operator to explicitly show how it interpolates gradients toward a consensus direction and unified the notation throughout the paper. We provided ablation studies on hyperparameters such as $\mu$ and $\lambda$, showing the method is stable across $\lambda$ values and benefits from high momentum ($\mu=0.9$). Crucially, we differentiated RGB from RotoGrad, explaining that RGB operates efficiently in gradient space using scalar angles ($O(T^2 D)$) to preserve task-specific information, whereas RotoGrad is computationally heavier ($O(KD^3)$) and sacrifices local information.
>
> ## Response to Reviewer xkvf
>
> We provided empirical and statistical validation for our theoretical claims, specifically confirming that the “Optimal Gradient” behaviour described in Figure 1 occurs in practice. We showed that RGB maintains a better balance between conflict reduction and gradient proximity (MER Balance) compared to baselines, leading to lower condition numbers and smoother optimization trajectories on complex benchmarks like QM9. To address concerns about statistical robustness, we re-ran experiments for CelebA (including baselines) with multiple seeds, demonstrating that while variance is inherent to the non-convex landscape, RGB’s relative ranking remains consistently superior.
>
> ## Response to Reviewer vAwe
>
> We clarified our method’s novelty and complexity, emphasizing that RGB is a global Gradient-Manipulation (GM) method, distinct from weighting-based approaches (like CAGrad) and superior to pairwise GM methods (like PCGrad) due to its holistic conflict resolution while highly preserving task-specific information. We demonstrated that the method is robust to noise by using an Exponential Moving Average (EMA) as a low-pass filter for the consensus direction. Furthermore, a detailed complexity analysis was added, confirming that RGB matches standard baselines with $O(TD)$ space complexity and comparable training time ($\approx 12.5$ min/epoch) for QM9, while incurring zero additional cost during inference.
>
> We hope this summary helps clarify our contributions and the improvements made during the rebuttal period. Thank you.

---

### Meta-Review · Area_Chair_1MiU · 2026-01-07

**Summary:**

All reviewers agree RGB offers a principled, globally-coordinated gradient-rotation remedy for MTL conflicts and delivers strong numbers, especially on QM9.
Remaining doubts center on (i) limited novelty vs. RotoGrad/CAGrad/Nash-MTL, (ii) missing thorough ablation (λ, μ, T-scaling), (iii) unconvincing complexity analysis, (iv) single-seed CelebA, and (v) superficial visual/presentation issues.

**Reviewer Concerns:**

Addressed: mathematical clarity of rotation operator; EMA & λ ablations; multi-seed CelebA; complexity table; head-to-head vs. RotoGrad; visual samples; notation; proof sketches.
Still outstanding: fundamental novelty over existing rotation/weighting methods; detailed FLOPS/memory scaling curves for >40 tasks; broader visual-task superiority (NYUv2 only leads on segmentation).

**Reviewer Scores:**

nN2S 4 → 4
xkvf 6 → 6
vAwe 2 → 3

---

### Decision · Program_Chairs · 2026-01-26

Reject